# Host Protein Kinase Cα: The novel Mitogen Activated Protein Kinase (MAPK) specific scaffold regulating nuclear export of influenza virus ribonucleoprotein complexes

Indrani Das Jana[1], Soumik Dey[1], Manoj Si[1], Arunava Roy[2¤], Arindam Mondal[1]*

**1** Department of Bioscience and Biotechnology, Indian Institute of Technology Kharagpur, Kharagpur, West Bengal, India, **2** Department of Molecular Medicine, University of South Florida, Tampa, Florida, United States of America

¤ Current address: Department of Interdisciplinary Oncology, Louisiana State University Health Sciences Center (LSUHSC), New Orleans, Louisiana, United States of America

* arindam.mondal@iitkgp.ac.in

## Abstract

Host protein kinase C (PKC) isoforms are well known modulators of different steps of influenza virus replication cycle. PKCα was reported to activate the Rapidly Accelerated Fibrosarcoma (Raf)/ Mitogen-activated protein kinase kinase (MEK)/ Extracellular signal-regulated kinase (ERK)- mitogen-activated protein kinase (MAPK) pathway to promote nuclear export of influenza virus ribonucleoprotein complexes (RNPs). However, the molecular mechanism by which PKCα activates specific members of the MAPK cascade and thereby facilitate virus replication, has never been investigated. Here we unravel the novel role of PKCα as a MAPK-specific scaffold to bridge stable kinase-substrate interaction between ERK2 with influenza virus nucleoprotein NP, the major constituent of RNP. Using analogue sensitive kinase, we show that ERK2 can directly phosphorylate NP at specific serine-threonine residues, which promote vRNP nuclear export and are indispensable for virus propagation. PKCα not only activates MAPK cascade, but also participates in stable interactions with the upstream kinase MEK1, effector kinase ERK2, and the substrate NP, thereby forming a multiprotein complex that regulate ERK2 activation, substrate recognition and subsequent phosphorylation events. This multiprotein complex localizes in the nucleus early during infection but eventually moves into cytoplasm at later stages of the viral life cycle. Overexpression of a dominant negative variant of PKCα blocks this complex formation, vRNP export and progeny virus production, thereby establishing PKCα as a key regulator of influenza virus replication. In summary, our results advance the molecular level understanding of the cross-talk between PKCα and MAPK pathway supporting influenza A and B virus replication.

**Data availability statement:** All relevant data are within the manuscript and its Supporting Information files.

**Funding:** A.M. acknowledges the Core Research Grant (CRG) from Anusandhan National Research Foundation (ANRF)/ Science and Engineering Research Board (SERB-DST (CRG/2022/003628) for providing financial support. A.R. acknowledges support from Institutional Research Grant IRG-21-145-25 from the American Cancer Society (ACS). I.D.J. acknowledges ANRF/ SERB for the National PostDoctoral Fellowship (PDF/2023/000284). S.D. acknowledges the support of the Ministry of Education (MoE) for Prime Minister's Research Fellowship (PMRF) doctoral fellowship. M.S. acknowledges University Grant Commission (UGC) for his doctoral fellowship. The funders had no role in study design, data collection and interpretation, or the decision to submit the work for publication.

**Competing interests:** The authors have declared that no competing interests exist.

## Author summary

Kinases are key regulators of various signalling cascades. Kinase-substrate interactions are generally transient but can be stabilized by scaffolding or anchoring proteins. This study shows how influenza virus exploits host PKCα as a scaffold protein to activate other cellular kinases (MEK1 and ERK2) and thereby recruit them upon viral replication machinery (RNPs). This RNP associated multi-kinase complex facilitates ERK2 mediated phosphorylation of viral NP protein, which regulates RNP's transport across nuclear membrane and thereby control the production of new virion particles. This work elucidates an unconventional mechanism of kinase-substrate interaction which is critical for influenza virus replication and hence could be targeted to develop novel host directed anti-influenza therapy.

## Introduction

Influenza viruses cause respiratory illness leading to seasonal epidemics and sporadic pandemics with variable morbidity and mortality rates [1]. Rapid evolution through antigenic drifts and antigenic shifts results in the emergence of newer influenza virus strains and subtypes rendering considerable challenge to the existing healthcare infrastructure [2,3]. Hence there is a pressing need of characterizing conserved virus-host interactions that could be targeted to develop newer host directed therapeutics.

Ribonucleoprotein particles (RNPs) of influenza viruses execute viral gene transcription and genome replication within the host cell nucleus. Eight negative sense viral genomic RNA segments, individually gets enwrapped with nucleoprotein (NP) oligomers and associates with viral RNA polymerase (RdRp) constituted of PB1, PB2 and PA subunits, together forming the viral RNPs (vRNPs) [4–7]. Upon entry into the host cells, incoming vRNPs are actively transported to the nucleus within an hour post infection (hpi) [8–11]. In the nucleus, RNPs perform primary transcription early during infection [11,12]. At later stages, in conjunction with newly synthesized viral NP and RdRp proteins, RNPs execute viral genome replication and assemble progeny vRNPs. Newly assembled vRNPs are actively exported out of the nucleus within 5–6 hpi and trafficked to the plasma membrane to assemble into new virion particles [9,11,13,14]. Viral matrix protein (M1) and nuclear export protein (NEP) together recruit host chromosome region maintenance protein (CRM-1) to mediate nuclear export of newly synthesized vRNPs [11,15–20]. The nuclear import-export of influenza vRNPs are tightly regulated through dynamic phosphorylation of NP at multiple serine, threonine and tyrosine residues and blocking any of these phosphorylations attenuates virus replication in cell culture and in vivo [21–26]. Although the role of NP phosphorylation in vRNP's nuclear-cytoplasmic trafficking has been extensively studied, the host kinases responsible for NP phosphorylation and their spatiotemporal regulation are sparsely characterized [26,27].

The mitogen activated protein kinase (MAPK), Raf/ MEK/ ERK signalling has long been known to modulate influenza virus life cycle, specifically by regulating vRNP nuclear export [28–30]. Previous studies showed that membrane accumulation of viral hemagglutinin (HA) protein, during early and late phases of infection, leads to a biphasic activation of the MAPK pathway ensuring a timely export of vRNPs during the late phase of the virus life cycle [29]. Supporting this notion, specific inhibition of MEK or ERK blocks vRNP nuclear export and severely restricts influenza A and B virus infection in cells and in vivo [25,28,31–35]. Recently, ERK activated p90 ribosomal S6 kinase 1 (RSK1) has been implicated in NP phosphorylation at S269 and S392 residues and promoting vRNP's nuclear export [25]. However, the recombinant NP-S269A/S392A mutant viruses lacking RSK1 phospho-sites, replicated to a significant extent ($10^6$ $\log_{10}$ by 32 hpi; one $\log_{10}$ lesser than WT virus) [25], hence raising the possibility of involvement of additional host kinases that compensates for this dearth of RSK1 mediated phosphorylation in promoting vRNP nuclear export.

In addition to MAPKs, different isoforms of the host PKC family are also known to modulate influenza virus infection [36,37]. Binding of viral HA to the cell-surface receptor activates the PKC pathway [29]. PKCδ has been shown to phospho-regulate NP oligomerization and assembly of the progeny vRNPs during genomic RNA replication [36]. The PKCδ can also phosphorylate RdRp subunits, which has been predicted to regulate viral transcription [38]. In contrast, PKCα activates the MEK/ ERK pathway in Ras independent but Raf dependent manner and thereby facilitates nuclear export of newly assembled vRNPs [28,29]. Pan PKC activators and inhibitors promote exclusive cytoplasmic or nuclear localization of NP/ RNPs respectively [13]. These data indicate an extensive cross-talk between PKCα and MAPK pathway moderating vRNP export, although the underlying molecular mechanism has never been explored. Whether PKCα acts just as a passive activator of the MAPK cascade, or plays a direct role in NP phosphorylation and vRNP trafficking across nuclear membrane is yet to be elucidated.

In this study, we present a detailed investigation to elucidate how PKCα modulates the MAPK pathway and acts as a master regulator of MEK-ERK mediated NP phosphorylation and vRNP nuclear export. We show that ERK2 acts as the effector kinase that directly phosphorylates influenza A and B virus NP proteins in vitro and using an analogue sensitive ERK2, we confirm this observation in the context of A/WSN/1933 (H1N1) infection. This ERK2 mediated NP phosphorylation promotes vRNP nuclear export and recombinant viruses lacking these ERK2 phospho-sites showed slower replication kinetics compared to the wild type (WT) viruses. Importantly, PKCα acts as a MAPK specific scaffold that bridges a stable interaction between MEK1, ERK2 and viral NP/RNPs, thereby ensuring robust activation and high substrate specificity of the MAPK cascade. This multi kinase-RNP complex localise within the nucleus, early during infection and dynamically regulate NP/ RNP trafficking across the nuclear membrane. Interference with the PKCα activity blocks vRNP export and attenuates virus replication, but mutant NP viruses lacking the ERK2 phospho-sites are resistant to this attenuation. Together our results elucidate a novel cross-talk between PKC-MAPK pathway which is utilized by influenza viruses to regulate the timely progression of infectious cycle.

## Results

### 1. PKCα activated ERK2 phosphorylates influenza virus NP

PKCα can activate the MAPK cascade by directly phosphorylating Raf-1 which in turn phosphorylates and activates its downstream kinases, including ERK2 (Fig 1A) [39]. Previous report suggested ERK2's involvement in NP phosphorylation [28], although no concrete evidence is available till date. To investigate the cross-talk between PKCα and ERK2 that may regulate NP phosphorylation, we overexpressed influenza A/ WSN/1933 (H1N1) NP and host ERK2 in human embryonic kidney (HEK) 293T cells and stimulated them using a pan PKC activator, phorbol 12-myristate 13-acetate (PMA). We showed earlier that PMA stimulation leads to NP hyperphosphorylation, leading to the appearance of a slower migrating band in Sodium dodecyl sulfate polyacrylamide gel electrophoresis (SDS-PAGE) [40]. Interestingly, overexpression of ERK2 followed by PMA treatment elevated NP hyperphosphorylation by 2.5 fold compared to the empty vector control (Fig 1B, lane 2 versus lane 4). This suggests a potential synergism between PKC and MAPK activities in NP phosphorylation.

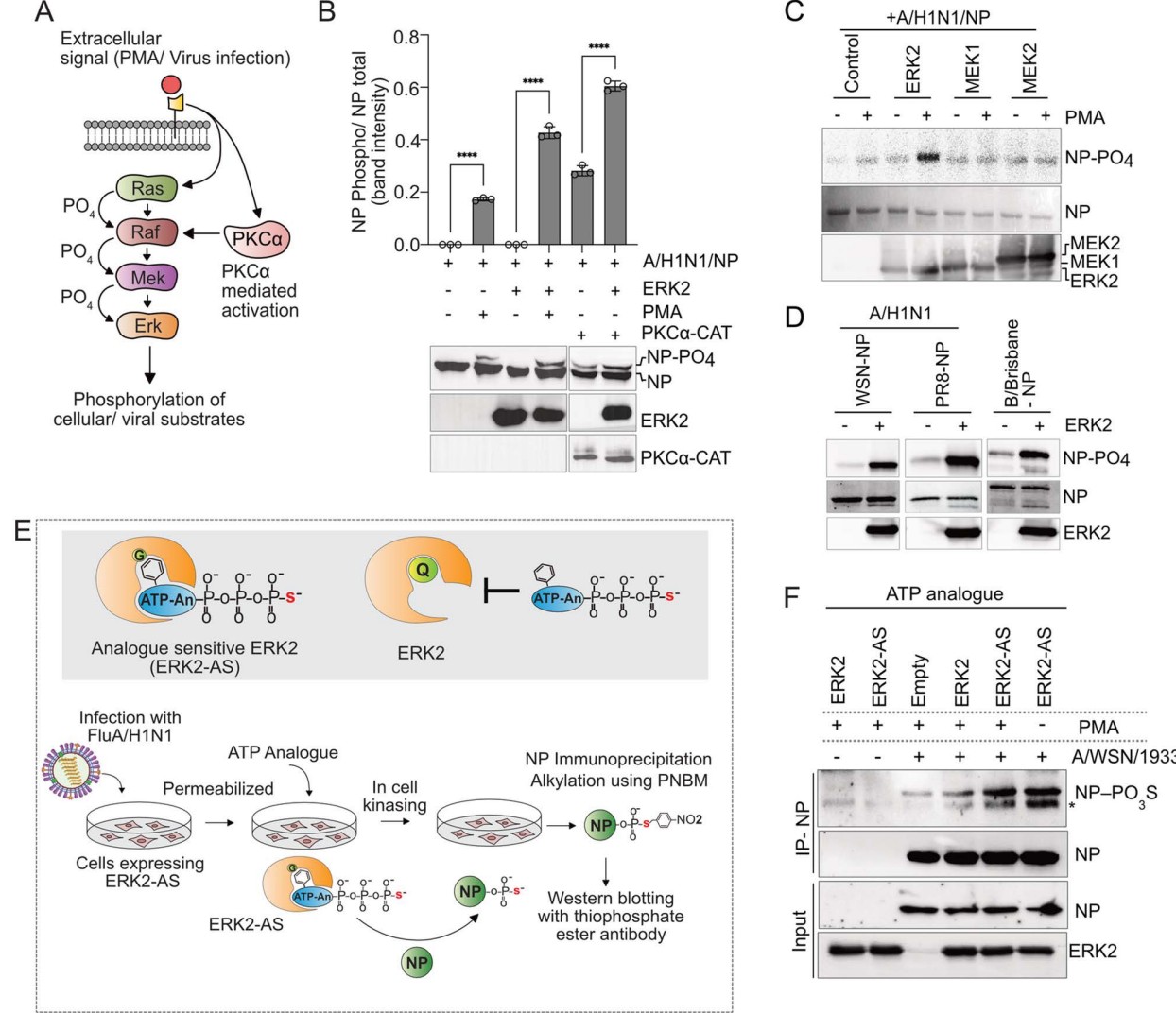

**Fig 1. ERK2 phosphorylates influenza A NP. (A)** Schematic representation of PKCα activated RAF-MEK-ERK pathway. **(B)** HEK293T cells over-expressing NP-V5, or co-expressing NP-V5 with ERK2-FLAG or PKCα-CAT-HA or both were treated with PMA or left untreated. NP, ERK2 and PKCα was detected by western blotting using V5, FLAG or HA specific antibodies. Band intensities of hyperphosphorylated and unphosphorylated NP were assessed from three independent experiments are plotted using image J software. **(C)** Cells overexpressing empty vector or FLAG tagged ERK2, MEK1 and MEK2 were stimulated with PMA and subjected to immunoprecipitation using FLAG antibody. Immuno-purified kinases were used to phosphorylate recombinant purified A//WSN/1933(H1N1) NP in vitro. Top panel: Phosphor-image showing NP phosphorylation, middle: Coomassie stained image of total NP protein; bottom panel: Western blot using FLAG antibody to show expression of kinases. **(D)** In vitro kinasing of recombinant A/WSN, A/PR8 and B/Brisbane NP using ERK2-FLAG purified from PMA stimulated cells. **(E)** Schematic representation of the mode of action of ERK2-AS and experimental design for the detection of ERK2-AS phosphorylated NP in infected cells. **(F)** Result of the experiment are presented in **(E)**. Asterisk in the panel F denotes non-specific host protein.

To further validate this, we used a constitutively active PKCα catalytic domain (PKCα-CAT) [41] that can trigger robust phosphorylation and hence activation of the ERK2 in cells (S1A Fig). Co-expression of PKCα-CAT, resulted in moderate hyperphosphorylation of NP, which gets significantly boosted in presence of ERK2 (Figs 1B, compare lane 5 and 6, and S1B). Considering that PKCα served as one of the upstream activators of the Raf/ MEK/ ERK cascade (Fig 1A), our data indicates that either ERK2 or one of its downstream effector kinases is phosphorylating NP. To test whether ERK2 can

directly phosphorylate NP, we performed an in vitro kinase assay where FLAG-tagged ERK2 was overexpressed and affinity purified from PMA stimulated HEK293T cells and was subsequently used to phosphorylate recombinant purified NP. We also purified MEK1 and MEK2 from PMA stimulated cells and tested their ability to phosphorylate NP in vitro. As shown (Fig 1C), active ERK2 directly phosphorylated A/WSN/1933 (H1N1) NP while MEK1 or MEK2 failed to do so. ERK2 also phosphorylates the influenza A/ PR/8/34 (H1N1) and the influenza B/ Brisbane/60/2008 NP proteins in vitro to an extent comparable to the A/ WSN/1933 (H1N1) NP (Fig 1D). These data suggest a direct kinase-substrate relationship between the host ERK2 and viral NP protein, which is largely conserved across influenza A and B viruses.

To validate the in vitro kinase assay result in the context of the virus life cycle, we adopted a chemical-genetic approach and employed the analogue sensitive ERK2 (ERK2-AS) to phosphorylate viral NP within the cell during the course of infection. The gatekeeper residue (glutamine, Q105) situated within the substrate binding pocket of the human ERK2 was mutated to glycine (G) [42] to make it compatible of utilizing the thio-adenosine triphosphate (ATP) analogue (Fig 1E, upper panel). The HEK293T cells either overexpressing ERK2 or ERK2-AS, were infected with A/WSN/1933 (H1N1) virus (MOI 0.01). At 12 hours post infection (hpi), the cells were stimulated with PMA, or left untreated, followed by in cell kinasing in presence of thio-ATP analogues. ERK2-AS phosphorylated NP should harbor thio-phosphate moiety conjugated at the phosphorylation sites, which upon alkylation using p-nitrobenzyle mesylate (PNBM) could be detected with thiophosphate ester specific antibody (method depicted in Fig 1E lower panel). As shown, ERK2-AS catalysed thio-phosphate conjugation to the viral NP protein with high efficiency in cells infected with influenza A virus and stimulated with PMA (Fig 1F lane 5). Additionally, virus infection alone (without PMA treatment) also triggered ERK2-AS mediated NP phosphorylation to the extent comparable to the PMA treated set (Fig 1F, lane 6), thus indicating robust activation of the MAPK pathway, specifically ERK2, during influenza virus infection. Notably, wildtype ERK2 catalysed thio-phosphate conjugation to NP comparable to the empty vector control (Fig 1F, lane 3 and 4), which might have resulted from the high abundance of the ATP analogue in cells during the in-cell kinasing reaction. Together, our data unambiguously established that influenza virus NP is a substrate for ERK2 mediated phosphorylation and suggested a role of PKCα in promoting this phosphorylation.

## 2. ERK2 phosphorylates specific serine threonine residues in NP

Next, we intend to identify the ERK2 phosphorylation sites within NP. Recombinant purified NP forms homo-oligomers (trimer/ tetramer) [43], thereby reducing the accessibility of a large number of phospho-sites, situated along its homotypic interaction interface, towards the kinases [36]. Hence, to nullify any effect of NP oligomerization towards ERK2 mediated phosphorylation, an obligatory monomeric NP mutant, NP-E339A, was used as a substrate for in vitro kinasing [43]. The monomeric NP-E339A appeared to be a superior substrate for ERK2, showing a 7.6 fold increase in phosphorylation compared to the wildtype NP (Fig 2A), which resembled our previous observation for PKCδ mediated NP phosphorylation [36]. To further confirm ERK2's specificity towards this monomeric NP-E339A, we used analogue sensitive ERK2 in the in vitro kinase assay. While both ERK2 as well as ERK2-AS could phosphorylate NP-E339A in presence of γ-thio ATP (Fig 2B, lane 2, 3), only ERK2-AS phosphorylated the monomeric NP variant in presence of γ-thio ATP analogue, (Fig 2B, lane 6), thereby confirming that it was ERK2 (and not any other kinase) that phosphorylated monomeric NP profusely in vitro. Subsequently, this in vitro phosphorylated NP-E339A was subjected to liquid chromatography coupled with tandem mass spectrometry (LC-MS/MS) leading to the identification of serine 450 and serine 473 as ERK2 phosphorylation sites (Figs 2C and S2 and S1-S3 Tables). The S473 residue is situated within a putative ERK2 target motif (Pro-X-S/T-Pro or S/T-Pro) while the S450 residue does not exhibit any such motif (Fig 2D). Structurally, S473 is situated in the body domain while the S450 is situated in the head domain, close to the homotypic interface of oligomeric NP (Fig 2D). Viral NP purified from the influenza A/WSN/1933 (H1N1) infected cells, when subjected to LC-MS/MS analysis, showed high abundance of phosphopeptides harbouring phosphorylation at S450, T472 and S473 amino acid residues of NP (Figs 2C and S3 and S4-S6 Tables), as also reported previously [40,44]. The adjacent position of T472 and S473 along with the preceding

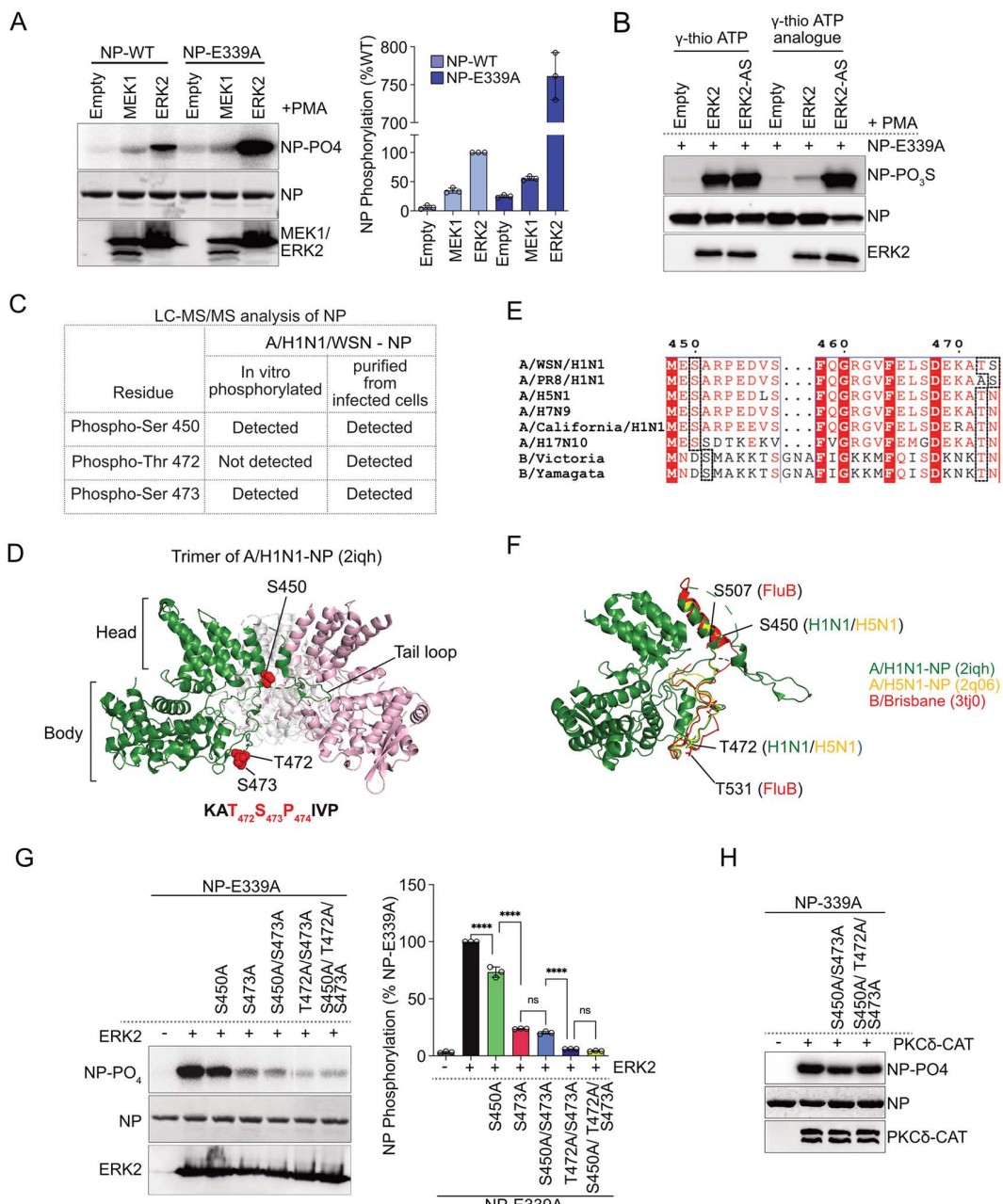

**Fig 2. ERK2 phosphorylates NP at specific serine/ threonine residues. (A)** WT NP or oligomerization defective NP-E339A proteins were in vitro phosphorylated using MEK1-FLAG or ERK2-FLAG. Band intensities from three independent experiments are measured using Image J software and plotted as relative percentage of WT NP phosphorylation by ERK2. **(B)** NP-E339A was phosphorylated in vitro with ERK2 or ERK2-AS in presence of γ-thio ATP or γ-thio-ATP analogue. **(C)** Results of the LC-MS/MS analysis identifying phosphorylation sites in recombinant purified NP-339A phosphorylated in vitro with ERK2 or NP purified from virus infected cells **(D)** Structure of trimeric A/WSN/1933 (H1N1) NP (PDB ID-2iqh) with individual monomers showed in different colors and highlighting the phosphorylation sites (identified through LC-MS/MS analysis) as red-spheres. The ERK2 consensus motif present in NP primary sequence is shown below. **(E)** Multiple sequence alignment of different influenza A and B virus NP proteins (from specific strains: A/WSN/1933(H1N1); A/PR8/1934(H1N1); A/Anhui/2005(H5N1); A/Anhui/2013(H7N9); A/California/2009(H1N1); A/Guatemala/060/2010(H17N10); B/Brisbane/60/2014; B/Wisconsin/15/2016) showing extent of conservation for the identified phosphosites (highlighted through dashed line). Detailed conservation analysis is included in S7 Table. **(F)** Alignment of the influenza A/WSN/1933 (H1N1) (2iqh), influenza A/HK/483/97 (H5N1) (2q06) and influenza B/Managua/4577.01/2008 (3tj0) NP monomeric units to show the structural conservation of the phosphosites. **(G)** NP-E339A harboring phosphonull alanine mutations were subjected to in vitro kinasing by ERK2. Extent of NP phosphorylation by ERK2 were measured from three independent experiments

and plotted using image **J**. Two way Anova was used to measure the statistical significance between the individual sets with P value (ns > 0.05; *P ≤ 0.05; **P ≤ 0.01; ***P ≤ 0.001) *P < 0.001 denotes statistically significant. **(H)** Phosphonull alanine mutations were subjected to in vitro kinasing by constitutively active PKCδ-CAT.

proline residue constitute the putative ERK2 phosphorylation site thus indicating that ERK2 may phosphorylate S450, T472 and S473 residues in infected cells. Interestingly, the NP S473 residue is least conserved with occurrence only in few specific strains of influenza A/H1N1, while the S450 and T472 remains largely conserved (both in primary sequence and in structure) across different influenza A subtypes and influenza B virus lineages (Fig 2E and 2F and S7 Table).

We mutated S450, T472 and S473 residues to phospho-null alanine, either individually or in combination, in the monomeric NP-E339A and assessed the propensity of these mutant NP proteins towards ERK2 mediated phosphorylation in vitro. As described earlier (Fig 2A), NP-E339A was used to nullify any potential effect of NP oligomerization upon the accessibility of the abovementioned phospho-sites towards the kinase. The phospho-null mutant NPs showed a varied degree of reduction in the ERK2 mediated phosphorylation (Fig 2G), however, no sensitivity was observed towards PKCδ (Fig 2H), a kinase that was previously reported to phosphorylate NP at serine 165 and serine 407 residues [33]. Individual mutation at S450A showed a moderate 27% decrease while S473A mutation showed a drastic 76% reduction in phosphorylation compared to the control NP-E339A. Interestingly, double mutant S450A/ S473A showed reduction similar to the S473A mutant (80% reduction), hence indicating that phosphorylation at S473 is essential for subsequent phosphorylation at S450. The double mutant harboring T472A/ S473A showed a drastic 94% reduction, which is comparable to the reduction observed for the triple mutant S450A/ T472A/ S473A (96% reduction). This data further indicates that T472 and S473 are redundant phosphorylation sites and phosphorylation at any one of these residues is a prerequisite for S450 phosphorylation. Together our data strongly establish the high specificity of ERK2 towards S450, T472 and S473 residues in NP.

### 3. ERK2 mediated NP phosphorylation is critical for virus replication

To investigate the role of ERK2 mediated NP phosphorylation in influenza virus life cycle, recombinant A/WSN/1933 (H1N1) viruses harboring NP-S450A, NP-S473A and NP-S450A/ S473A mutations were rescued. It is noteworthy that only S450 and S473 phosphorylation were identified through MS analysis of the NP that was in vitro phosphorylated by ERK2. Possibly, the adjacent T472 and S473 residues represent functionally redundant phosphorylation sites; hence only S473A mutation is included in virus rescue experiments. All the mutant viruses showed lower initial titre and smaller plaque size compared to the WT virus (Fig 3A) rescued in the same experiment. The mutant viruses showed slower replication kinetics in Madin Darby canine kidney (MDCK) cells (Fig 3B); the single mutants showed replication comparable to the WT virus at early time points (8–24 hours) but one log attenuation by 48–72 hours post infection (hpi). Viruses harboring the double mutant NP-S450A/S473A showed attenuation as early as 16 hours and the most drastic two log attenuation by 48 hpi onwards. Clearly, blocking the ERK2 mediated NP phosphorylation severely restricted virus replication thereby establishing the importance of these phosphorylation events in influenza virus life cycle.

To gain insight into the role of ERK2 in modulating specific steps of the influenza virus life cycle, we monitored viral RNA synthesis in cells infected either with WT or mutant viruses at different times post infection. Quantitative RT-PCR analysis performed using viral genomic RNA (vRNA) or viral mRNA specific primers showed that wild type and mutant viruses exhibited comparable vRNA and mRNA abundance as early as 1 hpi and 6 hpi. However, the mutant viruses showed attenuated RNA synthesis at later time points. The NP-S450A/S473A virus showed 50% reduction in vRNA synthesis at 12hpi, which further showed a gradual reduction to 95% by 48 hpi (Fig 3C and 3D). The NP-S450A and NP-S473A viruses however showed no or minor reduction till 24 hpi but then a drastic reduction to 16% and 18% respectively by 48 hpi (Fig 3C and 3D). The mRNA synthesis for WT and NP mutant viruses showed a trend similar to vRNA

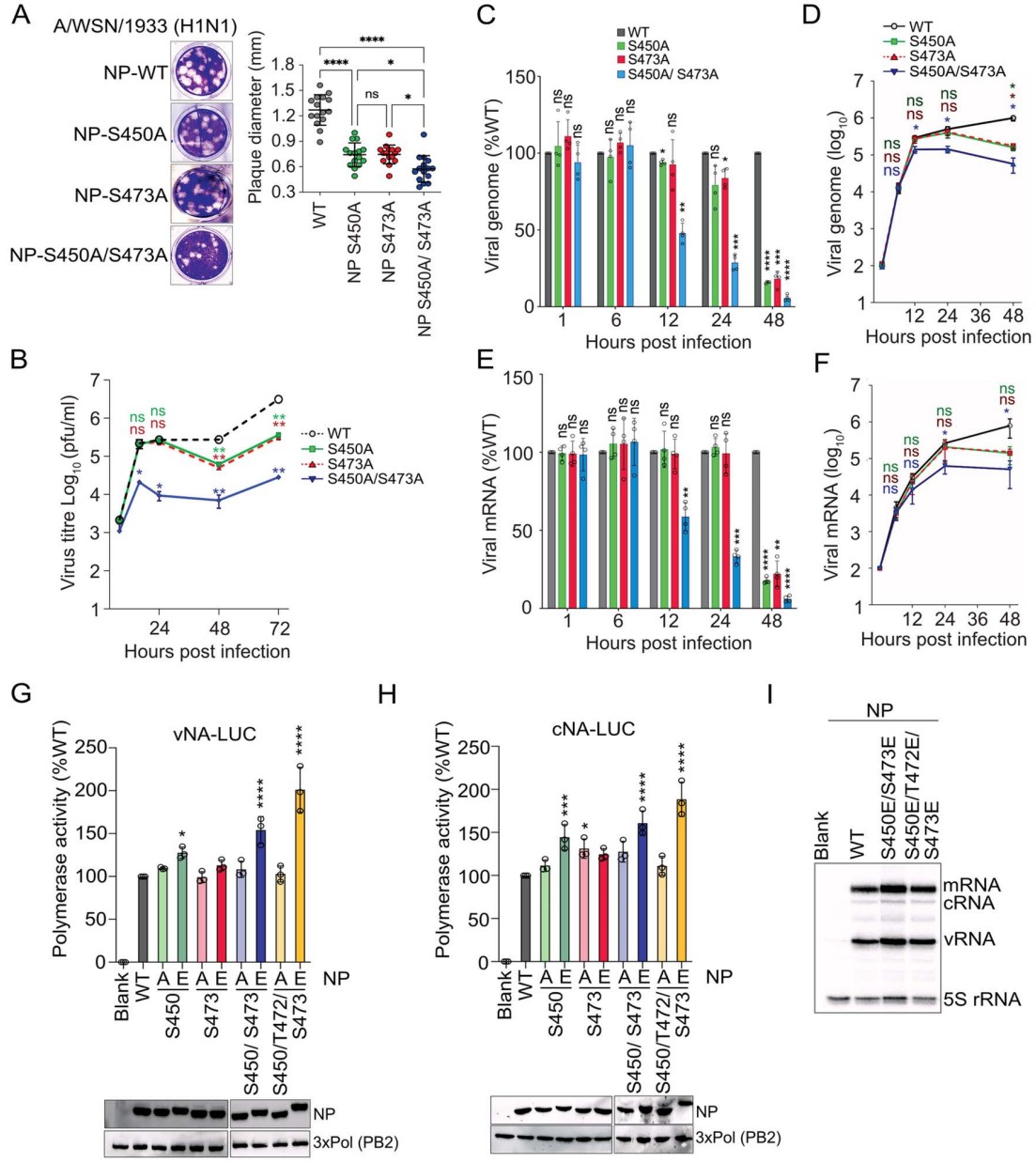

**Fig 3. ERK2 mediated phosphorylation is critical for virus replication. (A)** Plaque morphology and average plaque diameter of WT or recombinant NP mutant viruses. **(B)** Multicycle replication kinetics of the mutant NP viruses (MOI-0.01). Relative (C, E) and absolute (D, F) abundance of viral genome and viral mRNA at different times of post infection (MOI-0.05) were quantified using real time qRT-PCR. **(G, H)** RNP activity assay using negative and positive sense viral reporter RNA (vNA-LUC and cNA-LUC respectively) templates expressed along with RdRp subunits and WT and mutant NPs in HEK293T cells. (I) vRNPs were reconstituted in HEK293T cells with WT and mutant NP proteins as mentioned above and viral mRNA, cRNA and vRNA synthesis was measured using primer-extension analysis. Two way Anova was used to measure the statistical significance between the individual sets with P value (ns > 0.05; *P ≤ 0.05; **P ≤ 0.01; ***P ≤ 0.001).

synthesis for all the time points (Fig 3E and 3F). Considering that influenza virus completes a single infection cycle by 8 hours [45], this data suggests that the mutant viruses were not defective in RNA synthesis, rather one of the post viral RNA synthesis steps was blocked. To investigate this further, reporter viral RNPs were reconstituted in HEK293T cells to

monitor viral RNA synthesis independent of other steps of the virus life cycle. Plasmids expressing viral reporter RNA template (either negative or positive sense), RdRp subunits and NP protein were transfected in HEK293T cells and reporter activity was monitored as a proxy of viral RNA synthesis (Fig 3G and 3H). As evident, vRNP or cRNP reconstituted with phospho-null NP mutants (S450A, S473A, S450A/ S473A and S450A/ T472A/ S473A) barely showed any decrease in viral RNA synthesis, thereby suggesting the attenuation in virus replication was not due to a defect in RNA synthesis. Interestingly, RNPs reconstituted with multiple phosphomimetic glutamic acid substitutions in NP (NP-S450E/ S473E and NP-S450E/ T472E/ S473E) showed higher RNP activity (Fig 3G and 3H). This positive regulation of viral RNA synthesis was also observed in primer extension analysis. Viral mRNA, cRNA and vRNA synthesis was significantly boosted for RNPs reconstituted with specific phosphomimetic variants of NPs (Fig 3I). All the phosphonull and phosphomimetic NP mutant proteins showed expression levels comparable to the WT protein (Fig 3G and 3H, lower panel). These data together suggest that the ERK2 mediated NP phosphorylation is not essential for supporting viral RNA synthesis although phospho-NP molecules may serve as a superior substrate for the assembly of functional RNPs.

We have shown earlier that PKCδ mediated NP phosphorylation at its homotypic interface, regulates its oligomerization and assembly into viral RNPs [36,40]. The S450 residue is situated close to the homotypic interface of NP and participates in intermolecular interaction with the neighbouring protomer (Fig 2D and [43]). To test whether the ERK2 mediated phosphorylation at S450, T472 and S473 also regulate NP oligomerization, we utilized recombinant, purified WT and mutant NPs and subjected them to size exclusion chromatography. As evidenced (S4 Fig) WT and mutant NP proteins (S450A/ S473A, S450A/ T472A/ S473A, S450E/ S473E and S450E/ T472E/ S473E) showed similar oligomerization propensity with the NP phosphomimetic variants showing a minor increase in the abundance of the monomeric NP populations. This excludes any major role of ERK2 mediated phosphorylation in regulating NP oligomerization and RNP assembly.

## 4. ERK2 phosphorylation promotes nuclear export of NP and viral RNPs

Next, we investigated the potential role of ERK2 mediated phosphorylation in regulating NP's nuclear-cytoplasmic distribution. Individual NP mutants harboring either phosphonull or phosphomimetic mutations at S450 and S473 residues or both were transiently overexpressed in A549 cells and their subcellular localization was monitored using immunostaining with NP specific antibody. WT NP showed exclusive nuclear enrichment at early times of post transfection (12 hours) while demonstrating a mixed nuclear-cytoplasmic redistribution at a later time point (36 hours) (S5A Fig). NP mutants harboring phosphonull or phosphomimetic mutants at the ERK2 phospho-sites showed no difference in nuclear enrichment at 12 hours post transfection (h.p.t) (S5B Fig). At 24 h.p.t. both phospho-null and phosphomimetic NP mutants showed an intermediate phenotype (S5C Fig). However, NP phospho-null mutations showed low-moderate (for NP-S473A) to high (for NP-S450A/ S473A) nuclear retention compared to the WT at 36 h.p.t., hence suggesting a potential role of these phosphorylation sites in regulating NP nuclear export (Fig 4A). In line with this observation, phosphomimetic substitutions both at S450 and S473 residues (NP-S450E/ S473E) render the protein more efficient in nuclear export at 36 h.p.t thereby showing higher extent of cytoplasmic distribution with respect to the WT NP (Fig 4A).

In cells infected with WT A/WSN/1933 (H1N1) virus (MOI-5), NP showed nuclear accumulation as early as 2 hpi and mixed nuclear-cytoplasmic distribution by 8 hpi (Fig 4B). In contrast, viruses harboring phospho-null NP-S450A and NP-S472A mutations showed nuclear retention of NP even at 8 hpi and this alteration in nuclear-cytoplasmic distribution was more drastic for the virus harboring the double mutant NP-S450A/ S473A (Fig 4B). To establish that these phospho-null mutations not only alter NP localization but also interfere with the nuclear export of viral RNP complexes, we infected cells with either NP-WT or NP-S450A/ S473A viruses. Nuclear-cytoplasmic distribution of RNPs were monitored at 2 and 8 hpi by staining the cells with NP and PA (Fig 4C) or NP and PB2 (S6A Fig) specific antibodies. The double mutant virus showed significant retention of all the three RNP associated proteins in the nucleus at 8 hpi. Finally, we executed nuclear cytoplasmic fractionation of cells infected with WT or double mutant viruses, harvested at early and late times post infection (2 and 8 hpi). Relative abundance of viral genomic RNA (vRNA) and GAPDH in the respective sub-cellular

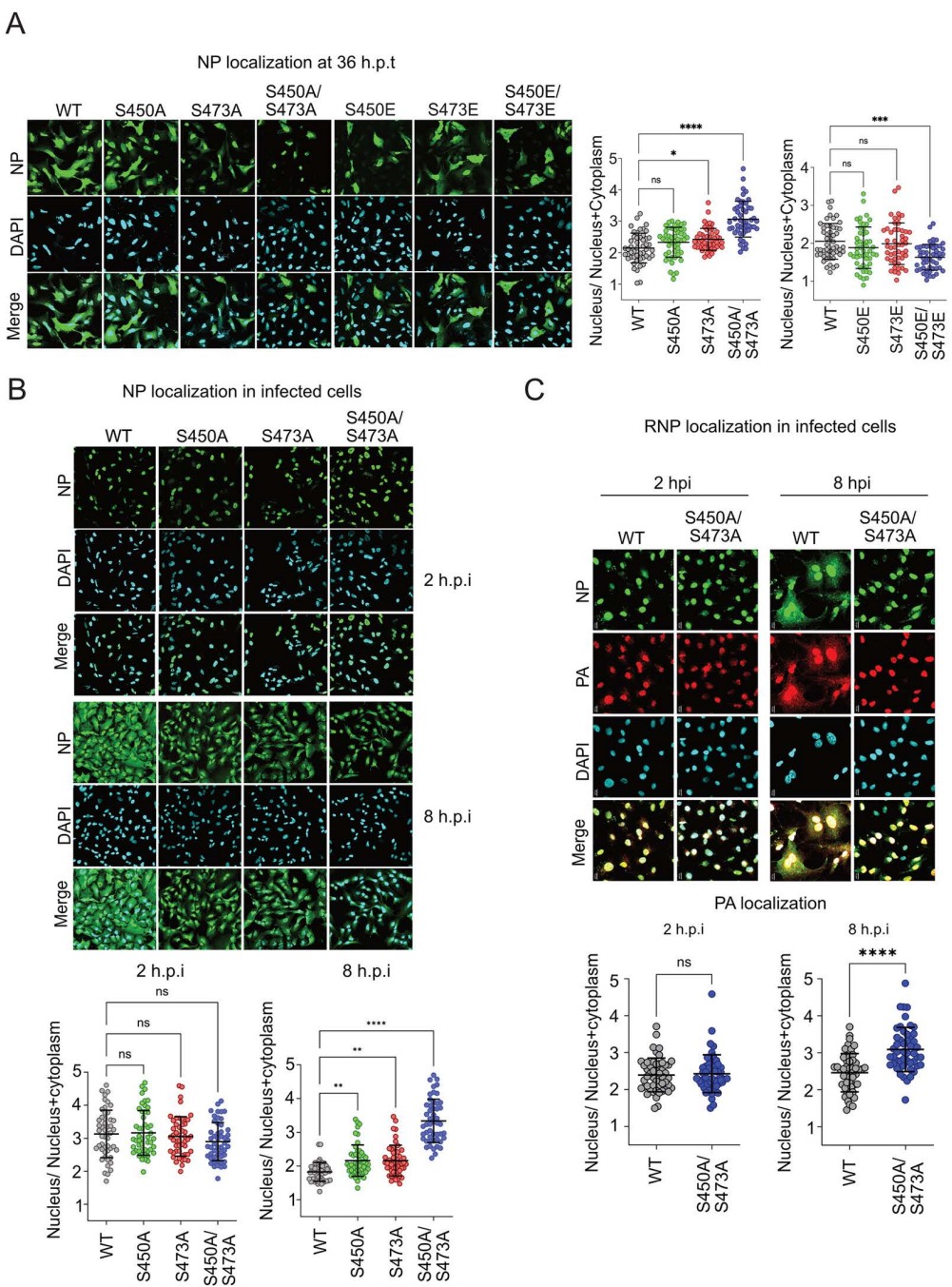

**Fig 4. ERK2 mediated phosphorylation promotes nuclear-cytoplasmic trafficking of NP/RNP complexes. (A)** WT and mutant NP proteins expressed in A549 cells were fixed at 12, 24 and 36 h.p.t, immuno-stained with anti-NP antibody and imaged using a confocal microscope. 12 and 24 h.p.t data presented in S5 Fig. **(B)** A549 cells infected with WT or mutant viruses (MOI:5). At 2 and 8 hpi cells were fixed, immuno-stained using anti-RNP antibody (detects NP) and imaged using a confocal microscope. **(C)** WT and S450A/ S473A mutant viruses were used to infect A549 cells. RNPs were visualised at 2 and 8 hpi by staining with NP and PA specific antibodies. Image analysis: 50 cells from 5 different fields were analysed using image J software to present nuclear-cytoplasmic distributions of NP. Each image is a representative of three independent biological replicates. Two way Anova was used to measure the statistical significance between the individual sets with P value (ns > 0.05; *P ≤ 0.05; **P ≤ 0.01; ***P ≤ 0.001).

compartments were analysed using Real-time PCR (qRT-PCR). As evidenced (S6B Fig) vRNAs of the WT viruses were majorly localised into the nucleus at 2 hpi (nucleus: cytoplasm~3:1) and redistributed to the cytoplasm (nucleus: cyto-plasm ~1:2) at 8hpi. In contrast, vRNAs of the double mutant virus were mostly retained within the nucleus throughout the time course of infection (nucleus: cytoplasm~2.5:1). GAPDH mRNA was prevalently localised in the cytoplasm irrespec-tive of infection, thus validating the sub cellular fractionation procedure. These data together demonstrate that ERK2 mediated NP phosphorylation is critical for promoting the nuclear export of the newly assembled vRNP complexes at the late stages of infection.

## 5. Active PKC-α scaffolds a stable interaction between NP, MEK1 and ERK2

So far we have characterised the role of ERK2 mediated NP phosphorylation in regulating RNP nuclear export and virus replication. Next, we aimed to characterize the molecular interaction between ERK2 and its substrate NP, which may facili-tate NP phosphorylation at specific sites mentioned above. While kinase substrate interactions could be transient, active ERK2 is known to participate in stable interaction with its substrates through recruitment of two docking domains [46]. Additionally, a number of scaffold proteins mediate interactions between ERK with its substrate along with the upstream kinases like MEK and Raf [47]. ERK2-NP interaction was investigated by co-expressing these two proteins followed by immunoprecipitating ERK2 and monitoring the co-precipitation of NP. No co-precipitation and hence no interaction were observed between the two proteins (Fig 5A lane 2). Considering that ERK2 activation may facilitate its interaction with NP, we included the constitutively active PKCα-CAT in this experiment, which is shown to activate ERK2 (S1A Fig) and trigger NP hyperphosphorylation (Fig 1B). Interestingly, the PKCα-CAT mediated stable and specific co-precipitation of NP with ERK2 (Figs 5A, lane 3, and S7). Additionally, ERK2 can also co-precipitate reconstituted viral RNPs but only in the context of PKCα-CAT overexpression (Fig 5A, lane 5). Interestingly, in both cases, the hyperphosphorylated form of NP was specifically enriched through ERK2 co-precipitation, hence suggesting that ERK2-NP interaction facilitates NP/ RNP phosphorylation. Apart from A/WSN/1933 (H1N1) NP, A/PR/8/34 (H1N1) and B/Brisbane/60/2008 NP, which were shown to get phosphorylated by ERK2 (Fig 1D), also co-precipitated with ERK2 in presence of PKCα-CAT (Fig 5B), thereby suggesting conserved nature of this kinase-substrate interaction in different influenza viruses. To test if MEK, the upstream kinase of ERK2, can also interact with NP, co-immunoprecipitation assay was performed with MEK1 or MEK2 co-expressed with NP either in presence or absence of the PKCα-CAT. Similar to ERK2, MEK1 but not MEK2 interacted with NP in a PKCα dependent manner (Fig 5C). Conversely, NP can also co-precipitate ERK2 and MEK1 in the reverse-coimmunoprecipitation assay, but only in the presence of PKCα-CAT and the extent of co-precipitation of MEK1 is much higher than that of ERK2 (Fig 5D). Notably, co-expression of ERK2/ MEK1 and PKCα-CAT significantly boosted NP expression levels (Fig 5A and 5C, NP expression blots, compare lane 2 & 3) which was not observed for co-expression of MEK2 and PKCα-CAT (Fig 5C, NP expression blots, compare lane 4 & 5), possibly because MEK2 was unable to partic-ipate in interaction with NP. PKCδ is known to interact with NP/ RNPs through the intermediacy of the PB2 subunit of the viral RdRp [36]. Considering its close association with viral RNPs, we asked if PKCδ can also mediate NP's interaction with ERK2. ERK2-NP co-immunoprecipitation was performed in presence or absence of constitutively active catalytic domains of this PKC isoforms. We observed that PKCα-CAT alone and not PKCδ-CAT facilitated coprecipitation of NP with ERK2 (Fig 5E), thereby establishing the exclusivity of PKCα in mediating this interaction. Together these data sug-gested that PKCα can mediate a stable interaction between ERK2 and MEK1 with NP (Fig 5F).

There could be two different models supporting PKCα mediating NP's interaction with MEK1 and ERK2. First, PKCα can activate the MEK1/ ERK2 cascade and hence facilitate ERK2 mediated NP phosphorylation. Second, PKCα may directly interact with MEK1, ERK2 and NP to act as a scaffold protein, thereby leading to the robust activation and high specificity of the MAPK cascade towards its substrate, NP. In order to test these hypotheses, we utilised a catalytically inactive version of PKCα-CAT (designated here as the dominant negative: PKCα-DN; reason described later) and tested its ability to mediate ERK2-NP interaction. The PKCα-DN failed to facilitate NP coprecipitation with ERK2 (Fig 6A, lane

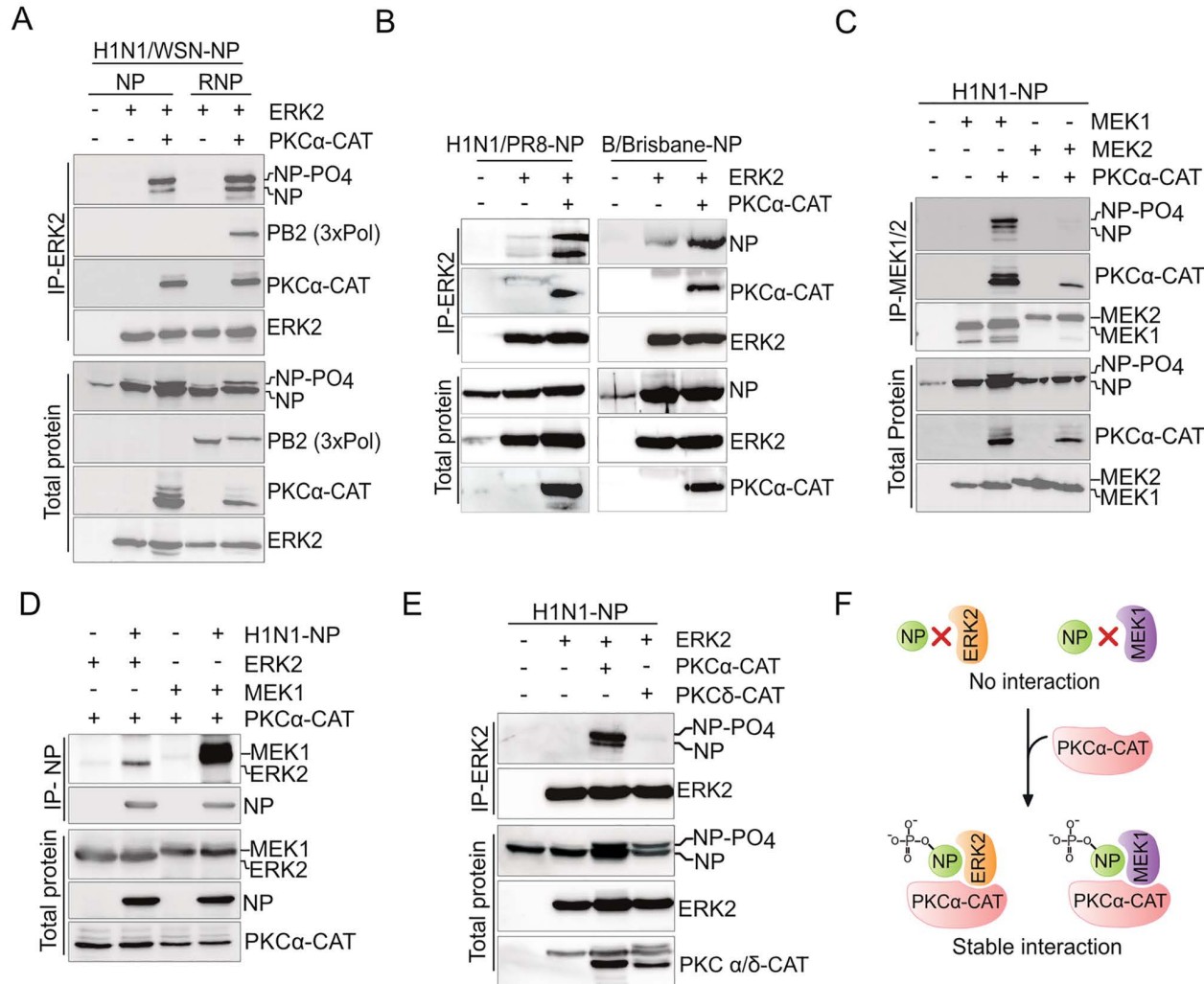

**Fig 5. PKC-α mediates a stable interaction between viral NP/ RNP with MEK1 and ERK2. (A)** Co-immunoprecipitation of A/WSN/1933(H1N1) NP/ RNP with ERK2. Cells were transiently transfected to overexpress NP-V5 or to reconstitute complete RNP complexes either in the context of ERK2-FLAG or ERK2-FLAG and PKCα-CAT-HA overexpression and immunoprecipitated using FLAG antibody. Co-precipitation of NP was observed using V5 antibody. **(B)** Co-immunoprecipitation of A/PR8/34(H1N1) NP and B/Brisbane/60/2008 NP with ERK2 either in presence or absence of PKCα-CAT as mentioned in **(A) (C)** Co-IP of NP with MEK1/2-FLAG in cells overexpressing A/WSN/1933(H1N1) NP either with MEK1/2 or MEK1/2 and PKCα-CAT. **(D)** Reverse Co-IP showing co-precipitation of ERK2-FLAG and MEK1-FLAG with immunoprecipitated NP either in presence or absence of PKCα-CAT. **(E)** Co-IP of A/WSN/1933(H1N1) NP with ERK2 either in absence or presence of PKCα-CAT or PKCδ-CAT. **(F)** Cartoon depicting PKCα-CAT mediated stable interaction between A/WSN/1933(H1N1) NP with MEK1 and ERK2.

4), hence suggesting that active PKCα is essential for supporting ERK2-NP/ RNP complex formation. Subsequently, we generated a constitutively active ERK2 (ERK2-CA) harboring L75P/S153D/D321N mutations [48] and tested its ability to interact with NP in presence and absence of PKCα (Fig 6B). Although ERK2-CA triggered robust NP hyperphosphorylation (as evident from total protein blot for NP), but it failed to directly interact with NP in the absence of PKCα. However, in presence of PKCα-CAT, ERK2-CA showed higher extent of NP co-precipitation compared to the WT ERK2 (Fig 6B, lane 5). It is evident that PKCα-CAT not only activates the MEK1-ERK2 cascade but also physically bridges the interaction of these kinases with NP. To validate this model further, interaction ability of PKCα-CAT along with MEK1, ERK2 and NP was investigated using co-immunoprecipitation analysis. PKCα-CAT co-precipitated both ERK2 and MEK1 (Fig 6C),

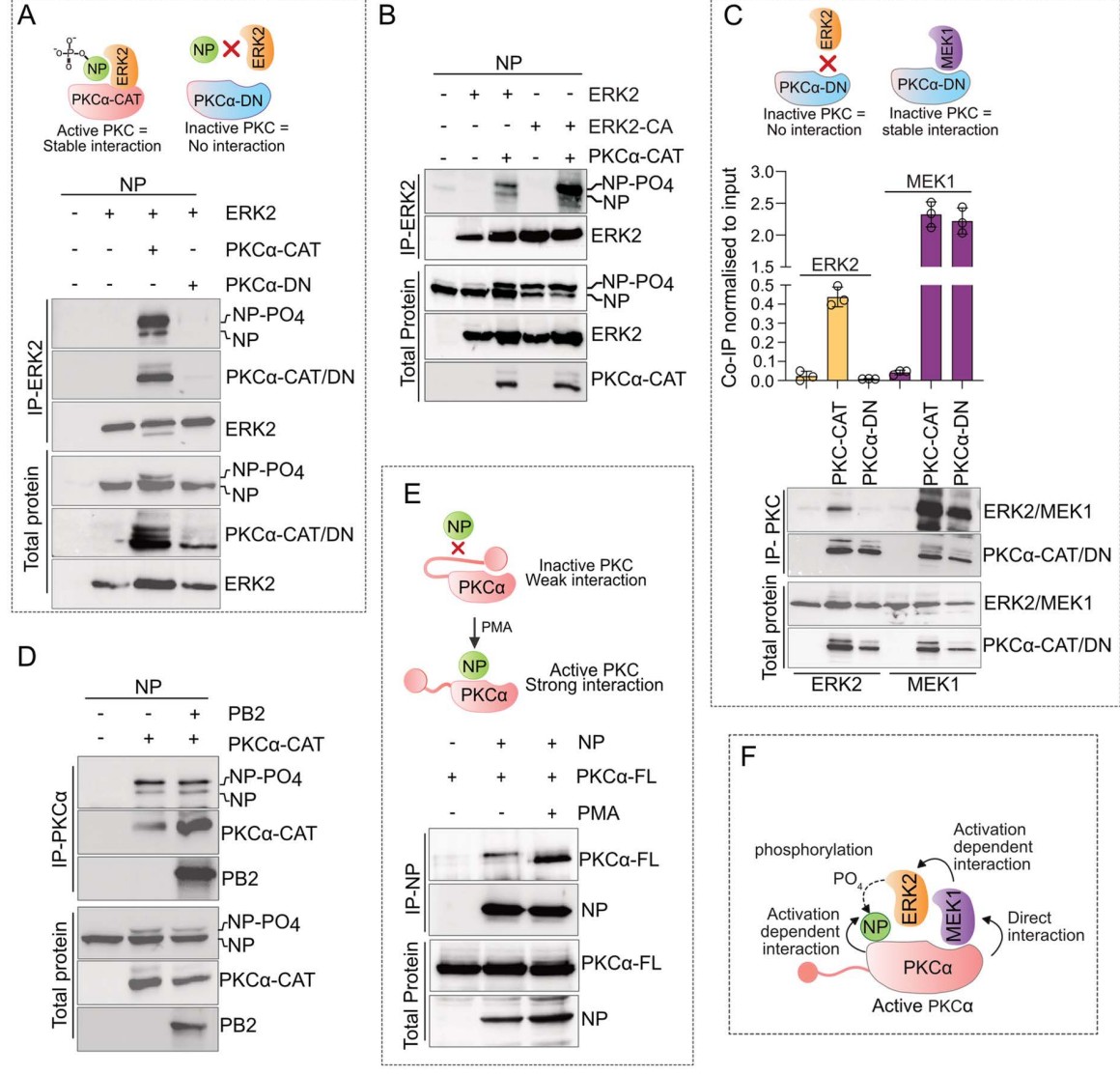

**Fig 6. PKC-α scaffolds a stable interaction between NP with MEK1 and ERK2. (A)** Co-immunoprecipitation of NP-V5 with ERK2-FLAG either in the context of PKCα-CAT-HA or PKCα-DN-HA overexpression. Cartoon depicting the results (upper panel) **(B)** Co-IP of NP with ERK2-FLAG or constitutively active ERK2-FLAG (ERK2-CA) in presence or absence of PKCα-CAT-HA. **(C)** Co-IP of ERK2-FLAG and MEK1-FLAG with PKCα-CAT-HA and PKCα-DN-HA. Densitometric analysis from three independent experiments is presented. Cartoon depicting the results (upper panel). **(D)** Co-IP of NP with PKCα-CAT either in the presence or absence of PB2. **(E)** Co-IP of full length PKCα (PKCα-FL) with NP overexpressed in cells stimulated with PMA or with vehicle control. Schematic representation of the result (upper panel). **(K)** Model describing the scaffolding role of active PKCα. All experiments are performed with A/WSN/1933(H1N1) NP.

and the extent of co-precipitation was 5 fold higher for the latter one. Thus, active PKCα interacts with MEK1 and ERK2 and shows stronger affinity for the former. This explains why MEK1 showed higher co-precipitation with NP than ERK2 in presence of PKCα-CAT (Fig 5D). Interestingly, the catalytically inactive PKCα (PKCα-DN) retained its ability to interact with MEK1 but failed to co-precipitate ERK2 (Fig 6C), thereby explaining why active PKCα is indispensable for mediating ERK2-NP interaction (Fig 6A). PKCα-CAT also interacted with NP when co-expressed and unlike PKC-δ, this interaction is independent of viral PB2 protein (Fig 6D; [36]).

Finally, we checked if the full length PKCα (PKCα-FL) could interact with influenza virus NP protein. As shown (Fig 6E) PKCα-FL, which usually remains in the inactive conformation [45], was able to co-precipitate low levels of NP but the co-precipitation was significantly increased upon PMA stimulation. Together, our data show that PKCα, in its active form, physically interacts with NP, MEK1 and ERK2 to form a multi-protein complex (Fig 6F) which is responsible for ERK2 mediated direct phosphorylation of NP. This also established PKCα as a scaffold protein for the MEK/ERK pathway.

## 6. The NP associated multi-kinase complex shows a dynamic nuclear-cytoplasmic accumulation at different stages of infection

Next, we aimed to gather evidence in support of stable association between the host kinases and NP during the course of infection. Components of the MEK-ERK cascade localizes within the cytoplasm in the resting cells but can be imported into the nucleus upon activation by extracellular stimulations [25,29,39]. Interestingly, influenza virus infection leads to the activation of ERK2 as early as 1–2 hpi and then at later stages, 6–8 hpi (S8 Fig), which corroborates with previous reports [28,29]. Subsequently, we checked if the activation of MAPK pathway can also alter subcellular distribution of endogenous ERK2, MEK1 and PKCα during the course of influenza virus infection. A/WSN/1933 (H1N1) infected A549 cells showed significant nuclear enrichment of PKCα, ERK2 and MEK1 compared to the mock infected cells at 2 hpi but not at 8 hpi (Fig 7A-7C). Clearly, PKCα along with MEK and ERK colocalize with NP within the nucleus early during infection. To further investigate this, we performed super-resolution microscopy of the infected A549 cells, and observed punctate structures of NP/ RNP complexes in the nucleus at 2 hpi and both in nucleus and cytoplasm at 8 hpi (Fig 7D-7F). At both the time points, ERK, MEK and PKCα proteins individually showed extensive colocalization with the punctuated structures of influenza NP/ RNP complexes at a close proximity of 50–60 nM approximately (Figs 7D-7F and S9 Table and S8 Table). Additionally, the majority of the co-localization was observed within the nucleus early during infection and in the cytoplasm during the late stage (Fig 7D-7F). These data suggest that the localization of ERK, MEK and PKCα dynamically changes with NP during the course of infection resulting in close association between these host proteins with viral RNPs, thereby facilitating interaction between them.

Encouraged by the colocalization data, we intend to establish that host kinases not only colocalize, but stably associate with each other and also with NP/ RNPs during influenza virus infection. Infection of A549 cells with A/WSN/1933 (H1N1) (MOI-10) followed by immunoprecipitation of endogenous PKCα showed a significant co-precipitation of both MEK1 and ERK2 proteins (Fig 8A). This provides evidence in support of the formation of PKCα-MEK1-ERK2 complex during influenza virus infection. To test, if this multi-kinase complex interacts with viral NP/ RNPs, A549 cells were either infected either with A/WSN/1933 (H1N1) or with B/Brisbane/60/2008 (MOI-10) viruses. NP/RNP complexes were immunoprecipitated and co-precipitation of endogenous PKCα and ERK2 was assessed using specific antibodies (Fig 8B and 8C). Both the upstream activator kinase, PKCα, and the downstream effector kinase, ERK2, were co-precipitated with NP/RNPs for both types of influenza viruses. Interestingly, the extent of co-precipitation was higher for PKCα than that of ERK2, possibly because of the direct interaction of NP with the former and indirect PKC bridged interaction for the later. This data along with the results presented in Figs 5 and 6, confirms the existence of the multi-kinase complex, constituted of PKCα, MEK1 and ERK2, which stably associates with different influenza virus RNPs during infection.

Finally, we probed the NP-ERK2 interaction to analyse the spatiotemporal dynamics of the NP associated multi-kinase complex formation during the course of influenza virus infection. We performed proximity ligation assay (PLA), wherein close association (within 40 nm) of the partner proteins appears as a fluorescent foci within the cell and the number of the foci in each cell could serve as a quantitative estimate of the corresponding protein-protein interaction (Fig 8D). A549 cells were either mock infected or infected with A/WSN/1933 (H1N1) virus, fixed at 2 hpi and 6 hpi and were subsequently subjected to PLA followed by detection of the fluorescent signal. Localization of viral NP was visualised using immunostaining post PLA treatment. Highly abundant fluorescent PLA dots were detected in the infected cells, both at 2 hpi and 6 hpi, when incubated with both anti-ERK2 rabbit and anti-RNP goat (majorly detects NP) antibodies; no signal was obtained for the mock infected

PLOS Pathogens

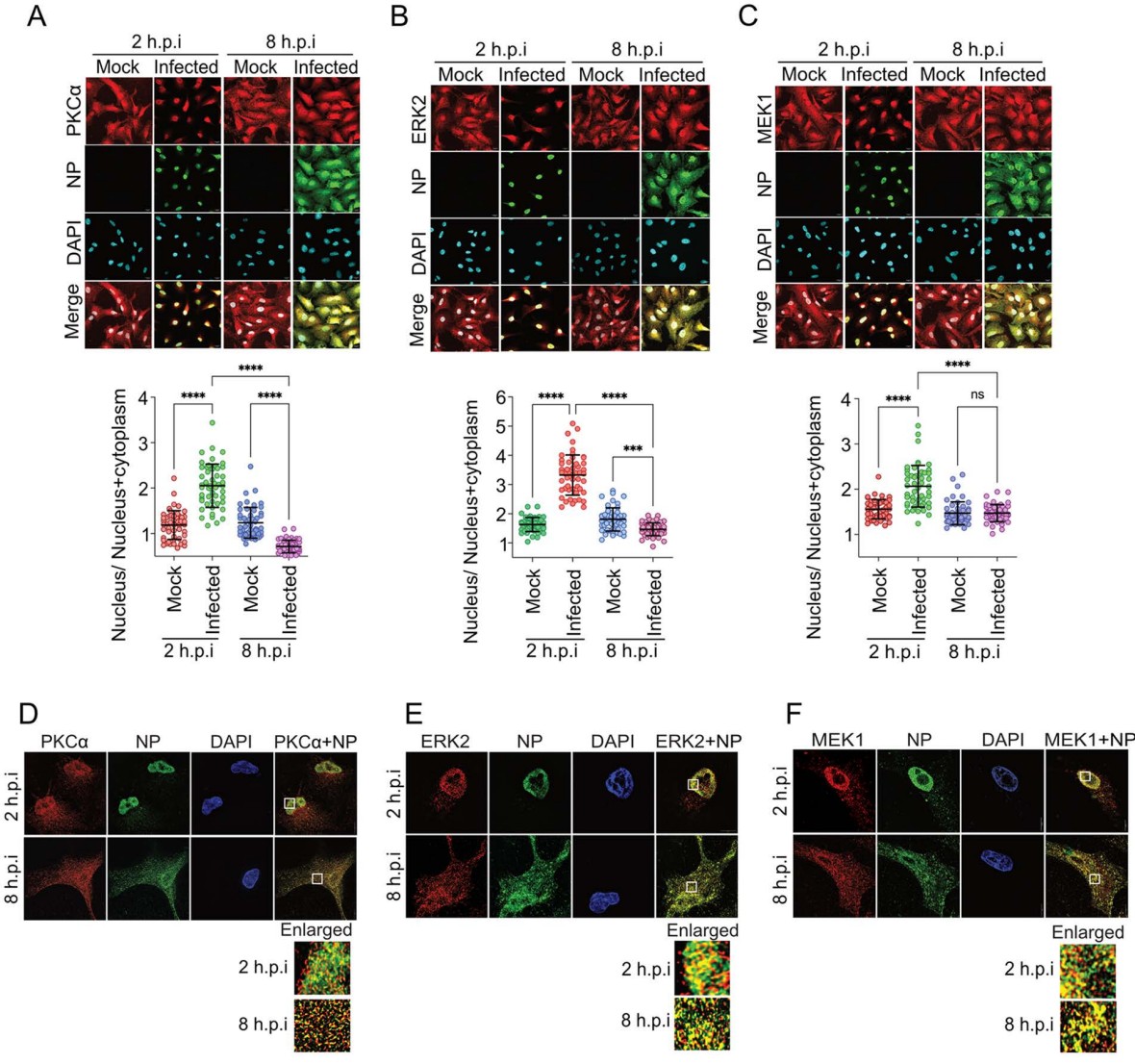

**Fig 7. PKCα, MEK and ERK's colocalize with NP/ RNPs in virus infected cells.** A549 cells infected with influenza A/WSN/1933(H1N1) virus (MOI-5) were fixed at 2 and 8 hours of post infection and immuno stained for NP and PKCα (A) or ERK (B) or MEK (C) with specific antibodies. Images from 5 different fields were analysed with image J software to present subcellular distribution of the proteins (lower panel). Each image is a representative of three independent biological replicates. Two way Anova was used to measure the statistical significance between the individual sets with P value (ns > 0.05; *P ≤ 0.05; **P ≤ 0.01; ***P ≤ 0.001). **(F)** Super-resolution micrographs of the infected A549 cells showing precise co-localization of viral NP with PKCα **(D)**, ERK (E) and MEK (F) at 2 and 8 hpi. White rectangle denotes the specific field which has been enlarged to show co-localization (extent of co-localization analysed using Image J software is presented in S9 Fig).

cells incubated with the same antibodies, or the infected IgG isotype treated control cells (Figs 8E, 8F, S10A, and S10B). This further confirmed a stable association between ERK2 and NP/RNP during the course of infection. Interestingly, the PLA dots were majorly located within the nucleus at 2 hpi but were distributed throughout the cytoplasm at 6 hpi, which perfectly coincides with the localization of NP within these virus infected cells (Fig 8E, 8F and 8G). Thus, the PLA result along with data presented so far collectively suggested that ERK2-NP complex formation happened within the nucleus, through the intermediacy of PKCα and MEK1, at early times post infection. The same complex gets exported out to the cytoplasm at late hours (6–8 hpi) post infection, which possibly mediates nuclear export of the newly assembled vRNP complexes.

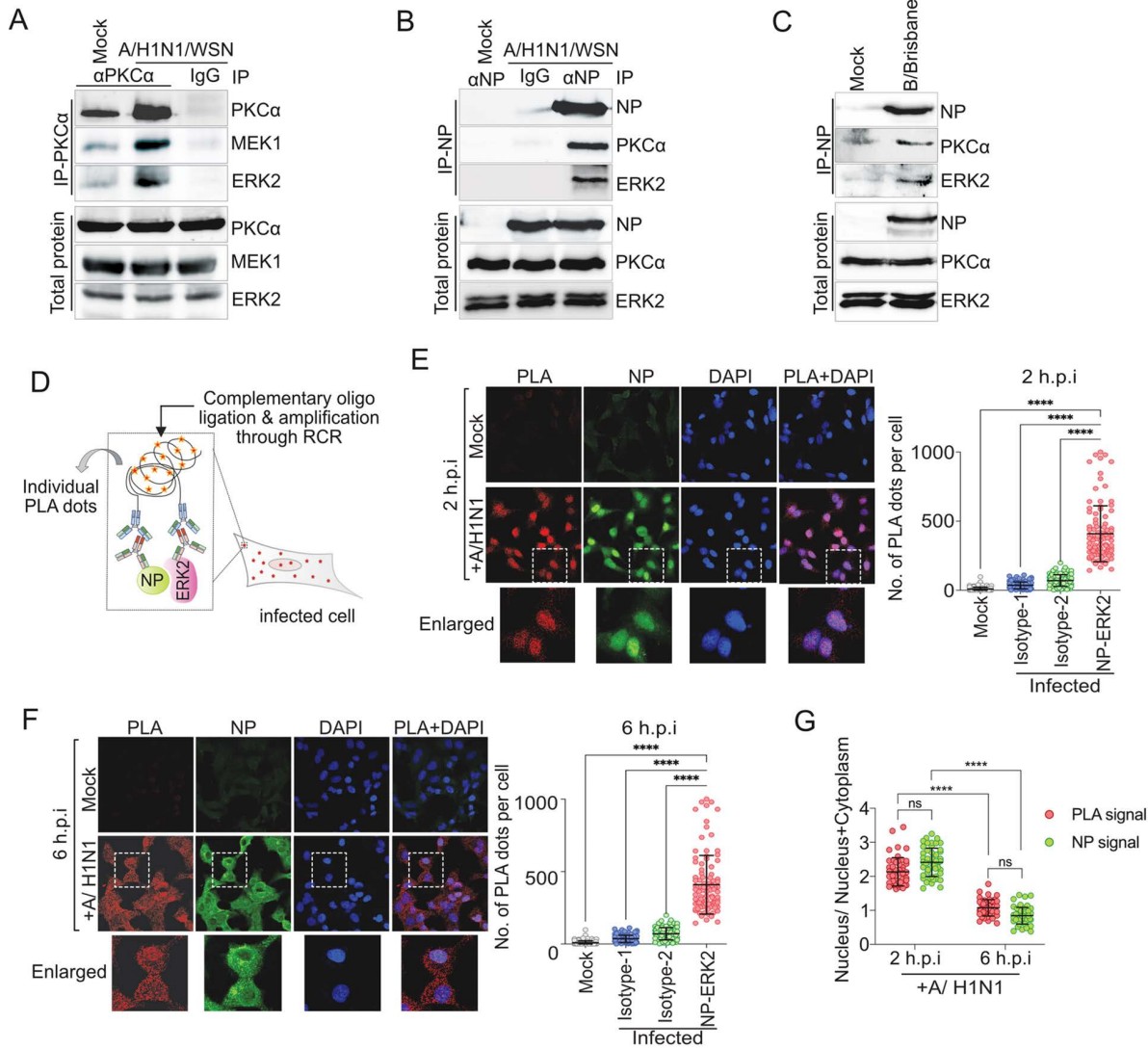

**Fig 8. Host kinases stably associate with NP and dynamically co-translocate between nucleus and cytoplasm. (A)** Immunoprecipitation of PKCα from A/WSN/1933(H1N1) infected (MOI-10) at 6 h.p.i; coprecipitation of endogenous MEK1 and ERK2 was observed using specific antibodies. Immunoprecipitation of NP from **(B)** A/WSN/1933(H1N1) virus or **(C)** B/Brisbane/60/2008 virus infected A549 cells (MOI-10) at 6 hpi using NP; coprecipitation of endogenous PKCα and ERK2 was observed using specific antibodies. **(D)** Schematic representation of proximity ligation assay. Cells infected with influenza A//WSN/1933(H1N1) virus or mock infected. At 2 hours (E) or 6 hours (F) of post infection, cells were subjected to PLA using anti-RNP (goat) and anti-ERK2 (rabbit) primary antibodies or with one of the primary antibodies along with the IgG isotype control for the other (full panel presented in S10 Fig). Post PLA, cells are stained with anti-NP antibody (mouse) and imaged. Images were analysed for the number of PLA dots per cell using Image J software and plotted for quantitative representation. **(G)** Subcellular distribution of PLA dots and NP were analysed using image J software. Two way Anova was used to measure the statistical significance between the individual sets with P value (ns > 0.05; *P ≤ 0.05; **P ≤ 0.01; ***P ≤ 0.001).

## 7. PKCα serves as the key regulator of vRNP export and influenza virus replication

Finally, we aimed to establish the role of PKCα mediated ERK2-NP complex formation in vRNP nuclear export and production of infectious virus particles. For this purpose, we used the catalytically inactive PKCα-DN, which can interact with MEK1 but failed to interact with ERK2 (Fig 6C) and hence was unable to bridge the ERK2-NP interaction (Fig 6A). We postulated that PKCα-DN, upon overexpression in cells, should sequester out endogenous MEK1 and interfere with

the PKCα-MEK1-ERK2-NP complex formation. Thus, PKCα-DN should act as dominant negative over the endogenous PKCα in terms of promoting ERK2 mediated NP phosphorylation and supporting the vRNP export at later times of infection. To validate this hypothesis, A549 cells overexpressing PKCα-DN were infected with influenza A/WSN/1933 (H1N1) and the NP/ RNP localization was monitored at 2 and 8 hpi (Fig 9A). As postulated, over expression of PKCα-DN severely restricted NP within the nucleus at 8 hpi. Importantly, the number of NP positive cells was also reduced by more than 50% with respect to the control set (Fig 9B), which indicated significant retardation of virus replication as early as 8 hpi. This reduction became even more prominent during multicycle replication where PKCα-DN overexpression resulted in two log reduction in virus titre at 12–36 hpi (Fig 9C). This established the critical role of PKCα in influenza A virus RNP export and virus replication, through mediating a stable ERK2-NP interaction and thereby promoting NP phosphorylation.

To draw a more direct correlation between PKCα's role in promoting influenza virus replication and ERK2 mediated NP phosphorylation, we generated PKCα deficient stable cell lines using shRNA mediated knockdown. Three different cell lines expressing shPKCα-1, shPKCα-2 and shPKCα-1 + 2 showed varied degrees of knockdown with the latter two achieving around 80% reduction in the PKCα level compared to the control cells (shNTC) (Fig 9D). A reduction in viral titer was observed in shPKCα-2 and shPKCα-1 + 2 cells at 12 hpi and more prominently at 24 hpi, compared to target shNTC cell line (Fig 9E-9J). For shPKCα-1, which showed slightly lesser efficacy of knockdown than the other two cell lines, lead to a significant reduction in virus titer only at 24 hpi. Interestingly, the double mutant NP-S450A/ S473A virus showed similar extent of replication in both the control and knockdown cells at both the times of post infection. Clearly, disruption of the ERK2 phosphorylation sites made the NP-S450A/S473A mutant virus insensitive towards the depletion in PKCα abundance in cell. This data unambiguously established the critical role of PKCα in regulating ERK2 mediated NP phosphorylation, thereby promoting vRNP export and ultimately facilitating the production of infectious virus particles.

## Discussion

Influenza virus RNP's optimal life span within the host cell nucleus and their timely export to the cytoplasm is essential for viral gene expression and genome replication and production of infectious virus particles. This timely coordinated nuclear-cytoplasmic shuttling of influenza virus RNPs is tightly regulated through MAPK mediated phosphorylation of NP at multiple serine, threonine residues [13,21–25]. In this study, we unravel the molecular mechanism by which PKCα activates the MEK-ERK pathway to regulate ERK2 mediated direct NP phosphorylation and promote vRNP's export from nucleus to the cytoplasm. We show that active PKCα scaffolds a stable tripartite interaction between MEK1, ERK2 and viral NP/ RNPs. This RNP associated multi-kinase complex is formed within the nucleus during early phase to promote NP phosphorylation and dynamically translocate to cytoplasm at later phase of infection.

Using analogue sensitive kinase, we established a direct kinase-substrate relationship between host ERK2 and influenza virus NP. ERK2 phosphorylates A/ WSN/1933 (H1N1) NP at Ser450, Thr472 and Ser473 residues wherein the latter two residues are situated within a consensus "Thr472-Ser473-Pro474" motif [49], a putative ERK2 phosphorylation site. Our data showed that phosphorylation at one of the consensus Thr472/Ser473 residues is pre-requisite for the phosphorylation at the non-consensus Ser450 residue. Interestingly, S450 and T472 phosphosites remain conserved across different influenza viruses suggestive of their essential role in supporting the virus replication cycle. In line with this observation, mutation of two out of the three phospho-acceptor sites (S450A/ S473A) makes the virus two log attenuated compared to the WT virus.

The MAPK, MEK-ERK pathway has long been implicated in RNP nuclear export [28–30]. Corroborating with this notion, phospho-null or phosphomimetic substitutions at the ERK2 phosphorylation sites imparted a negative or positive influence upon the RNP nuclear export. Additionally, phospho-mimetic substitutions at more than one phosphorylation sites boosted viral RNA synthesis, while the phosphonull substitutions failed to show any significant effect. The mutant viruses harboring phosphonull substitutions showed no defect in RNA synthesis during the early cycles of replication, rather demonstrated limited vRNP export and reduced virus titre, which indirectly impacted viral RNA synthesis at later infection cycles. These

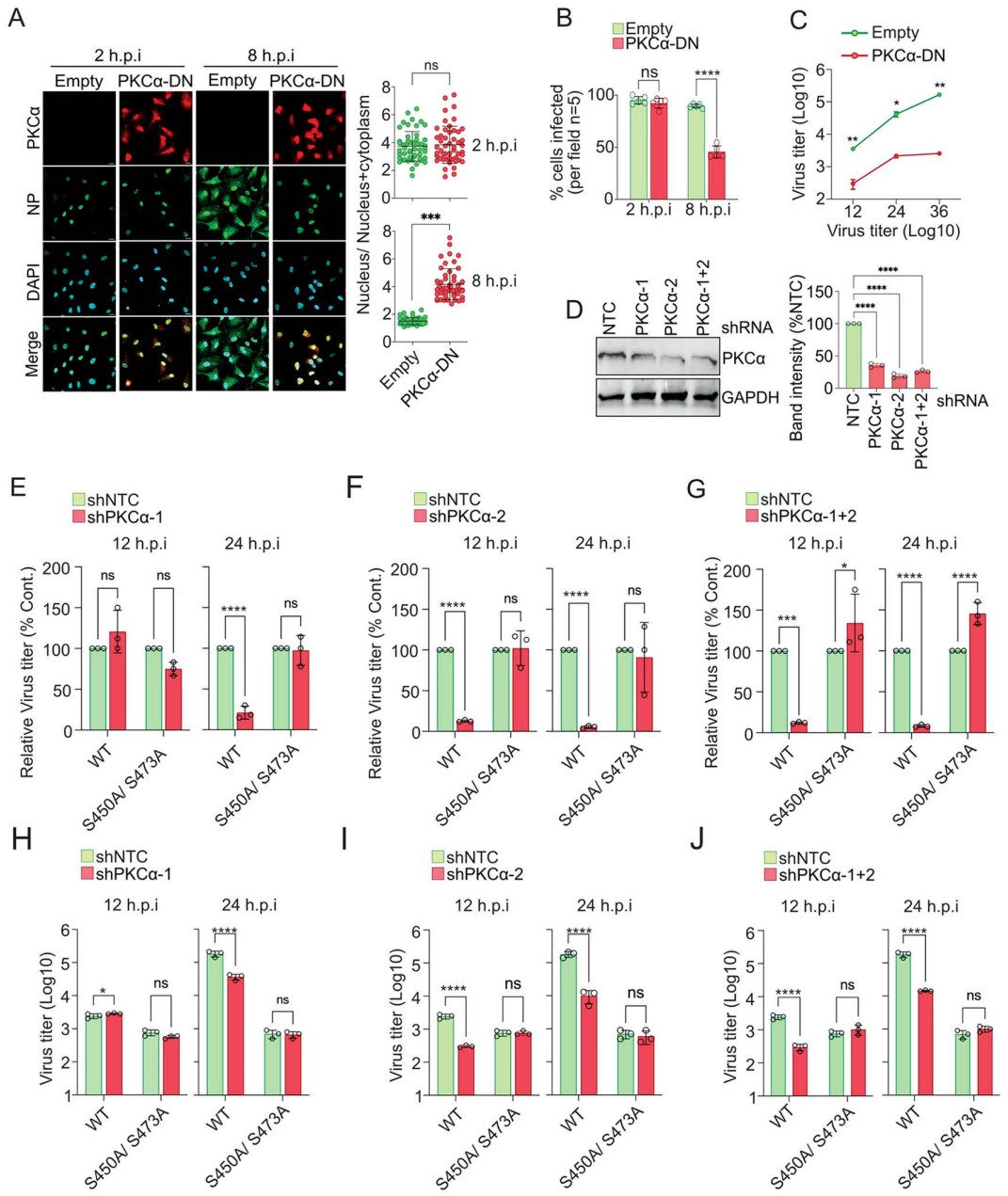

**Fig 9. PKCα as the key modulator of influenza virus replication. (A)** Cells overexpressing PKCα-CAT-DN or control cells were infected with A// WSN/1933(H1N1) virus followed by fixing and immunostaining at indicated times of post infection. **(B)** Images from five different fields were analyzed to measure the extent of infection using image J software. **(C)** Virus titers from PKCα-CAT-DN overexpressing or control cells were measured using plaque assay at different times of post infections as indicated. **(D)** A549 cells were transduced with lentiviruses expressing PKCα specific or non-target (NTC) shRNAs. Knockdown efficiency was tested by western blot analysis and PKCα band intensity from three different experiments are plotted using image J software. PKCα deficient cells (PKCα-KD1, PKCα-KD2, PKCα-KD1+2) were infected with WT or NP-S450A/ S473A mutant viruses and titer was measured at 12 and 24 hpi. Relative **(D-F)** and absolute **(G-I)** virus titers for both the viruses at two different time points are plotted. Each image is a representative of three independent biological replicates. Two way Anova was used to measure the statistical significance between the individual sets with P value (ns > 0.05; *P ≤ 0.05; **P ≤ 0.01; ***P ≤ 0.001).

data suggest that ERK2 mediated NP phosphorylation is essential for vRNP's transport from nucleus to the cytoplasm and hence production of progeny virions. It is noteworthy that, both ERK2 and its downstream kinase RSK1 phosphorylates distinct residues in NP to promote vRNP nuclear export [25]. Apparently, influenza viruses recruit multiple kinases at different tiers of the MAPK cascade to tightly regulate vRNP nuclear export that might be essential for efficient progression of the infection cycle.

ERK2's specificity towards its substrate can be determined through several factors including (i) cross-talk with other signalling cascades [50], (ii) dynamic alteration in subcellular localization [51]and (iii) interaction with scaffold proteins [52]. Our data suggests that influenza virus employs all of these strategies for ensuring ERK2 mediated specific phosphorylation of NP/ RNP complexes. First, we showed that PKCα, which is not an integral part of the MAPK cascade, activates ERK2 to phosphorylate NP (Fig 1). Second, influenza virus infection triggers PKCα, MEK1 and ERK2 translocation to the nucleus as early as 2hpi, coordinating with the localization of the newly synthesized NP proteins to the nucleus (Fig 7). Third, we present multiple evidences highlighting that PKCα acts as a scaffold to form a stable NP-PKC-MEK1-ERK2 complex (Figs 5–6, and 8). Interestingly, PKCα shows higher affinity towards MEK1 than ERK2 and the interaction with MEK1 is independent of its activation status. This is suggestive of a model (Fig 6F) where active PKCα interacts with NP and MEK1 directly, leading to MEK phosphorylation and subsequent activation. The PKC bound active MEK1 then interacts with ERK2 leading to its activation and also bringing it to the close proximity to its substrate, NP. In this regard it should be noted that the MEK1 is known to interact with ERK2, which helps ERK2 to transition to a conformation that is ideal for interaction with its substrate [53]. Thus, the scaffolding activity of PKCα creates a platform where the effector kinase ERK2 remains in close association with its activator kinase MEK1 and its substrate NP, thereby ensuring selective activation and high signalling specificity of the MAPK cascade. This model is indirectly supported by the fact that the catalytically inactive PKCα (PKCα-DN), which can co-precipitate MEK1 but not ERK2, failed to bridge NP-ERK2 interaction. In fact, the PKCα-DN, when over-expressed during the course of infection, imparts a dominant negative phenotype, possibly by sequestering out endogenous MEK1 from the multiprotein complex, thereby blocking the vRNP export and perturbing virus replication. This is further substantiated by the fact that PKCα knockdown significantly impacted replication of WT influenza virus but did not alter the replication of the mutant virus lacking the ERK2 phosphorylation sites in NP. Clearly, PKCα serves as a master regulator of influenza virus life cycle (i) by activating the MAPK pathway, (ii) mediating a stable interaction between different members of the MAPK cascade and influenza virus NP/ RNPs and (iii) triggering ERK2 mediated NP phosphorylation which ultimately promotes the vRNP nuclear export and progeny virus production.

The influenza virus activated ERK signalling pathway results in the activation of the downstream kinase, RSK1. This RSK1 mediated NP phosphorylation is known to regulate the vRNPs association with M1 and indirectly with CRM1, thereby promote the assembly of vRNP-nuclear export complex [25]. But, whether these phosphorylations occur prior to, during, or post RNP assembly, is not clear yet. Our immunofluorescence data suggests that activated PKCα, MEK1 and ERK2 translocate to the nucleus and co-localizes with NP as early as 2 hpi. PLA data also showed that ERK2 stably associate with NP during the same time within the nucleus and into the cytoplasm at later time points (6 hpi). Possibly, the newly synthesized NP molecules are quickly phosphorylated by ERK2 post nuclear translocation before participating in the RNP assembly process. The ERK2 phosphorylated NPs may serve as superior substrates for RNP assembly, as indicated by our RNP activity assay. Once assembled, these phosphorylated RNPs gets exported out to the cytoplasm through CRM1 dependent pathways. This might be the reason why the NP mutant viruses lacking the Rsk1 phosphorylation sites replicated to a significant extent [25], possibly facilitated by ERK2 mediated phosphorylation. It would be interesting to check if ERK1, which shares high sequence homology and functional redundancies with ERK2 may also participate in NP nuclear export and associates with RNPs during influenza virus infection. Additionally, the existence of ERK2- NP/ RNP complexes in the cytoplasm at the later stages of infection suggests that the RNP associated host kinase complex may also regulate post export trafficking and the virion assembly process which warrants further investigation.

In summary, we elucidate a novel mechanism highlighting the role of PKCα mediated activation of MEK-ERK pathway which involve formation of a multi-kinase complex that associate with influenza virus NP protein and phosphorylates it to promote vRNP nuclear export and production of infectious virion particles. The scaffolding activity of PKCα in mediating a MAPK-substrate interaction has never been reported for any other physiological processes, which suggests that influenza viruses utilize novel pathways to exploit multiple host kinases to facilitate its life cycle. Characterizing these interactions and understanding their spatiotemporal dynamics is critical for the detailed understanding of the virus-host kinase relationship. This will open up new scopes to target such pivotal virus-host interactions for developing new antiviral strategies.

## Materials and methods

### Cells, viruses, drugs and antibodies

Human lung epithelial cells (A549) (CCL-185), human embryonic kidney (HEK) 293T cells (CRL-3216) and Madin Darby canine kidney (MDCK) (CCL-34) cells were maintained in Dulbecco's modified Eagle's medium (DMEM) supplemented with 10% heat-inactivated fetal bovine serum (FBS) at 37°C and 5% $CO_2$ along with antibiotic-antimycotic containing 10,0000 unit/ml penicillin, 10,0000 μg/ml streptomycin antibiotics and 25 μg/mL of Amphotericin B (Gibco).

Influenza A virus strains, A/ Wilson-Smith/1933 (H1N1) encoding PB2 protein fused to a C-terminal Flag epitope tag (WSNPB2-FLAG) [54], A/Puerto Rico/8/1934 (H1N1) influenza A virus strain (A/PR/8/34) and influenza B (B/Brisbane/60/2008) viruses were used for infection in different cell lines.

Phorbol 12-myristate 13-acetate (PMA) (Sigma) was reconstituted in Dimethyl sulfoxide (DMSO) at a stock concentration of 100 μg/ml. Cells were stimulated with complete DMEM media containing 1.5 μg/ml of PMA in DMSO or solvent alone for 1 hour.

Antibodies used include anti-thiophosphate ester antibody ([51–8] (ab92570), Abcam), anti-HA (C29F4, Cell Signaling Technology, CST), anti-V5 (D3H8Q, CST), anti-NP (H16-L10-4R5,BEI Resources), anti-FLAG M2 (F1804-1MG,Sigma), anti-influenza A virus RNP (NR-3133, BEI Resources), MEK1/2 (L38C12, CST) and p44/42 MAPK (ERK1/2) (137F5, CST), PKCα (2056, CST), phospho- p44/42 MAPK (ERK1/2) (9101, CST), influenza B NP monoclonal antibody (GT371, Invitrogen), GAPDH (G8795, Sigma), monoclonal anti-influenza A virus Polymerase Basic subunit 2 (PB2), Clone F5-59 (NR-31694, BEI Resources) and monoclonal anti-influenza A virus Polymerase Acidic subunit (PA), Clone F1-2A5 (NR-31684, BEI Resources, NIAID).

### Plasmids

All virus related genes were derived from either influenza A/WSN/1933 (H1N1), influenza A/PR/8/34 (H1N1) or influenza B/Brisbane/60/2008 viruses. Plasmids encoding A/WSN/1933 (H1N1) NP- pCDNA6.2-NP-V5 and polymerase proteins- pCDNA3-PB2-3xFLAG (encoding a C-terminal FLAG tag), pCDNA3-PA and pCDNA3-PB1 were expressed in HEK293T cells. vNA-luc reporter plasmids encoding firefly luciferase in the negative sense flanked by untranslated regions (UTRs) from the NA gene was utilized [55]. Plasmids expressing the full-length isoform of PKCα (PKCα-FL, Addgene plasmids #21232), the catalytic domain (PKCα-CAT, Addgene plasmids #21234) [41] and PKCα-CAT-DN (with K368R mutation in the PKCα-CAT construct was used to generate catalytically inactive form) [56]. PKCδ-CAT was obtained from Addgene plasmids #16388. pET28a-NΔ7NP was used for the bacterial expression of WT A/WSN/1933 (H1N1)NP with a C-terminal His tag and seven amino acid deletion on the N-terminus, as described previously [40]. pET28aBNP was used for expressing B/Brisbane/60/2008 nucleoprotein (BNP) in bacteria. The pcDNA-ERK2-3xFLAG, pcDNA-MEK1-3xFLAG and pcDNA-MEK2-3xFLAG plasmid constructs were generated in the lab. Constitutively active form of ERK2 was generated by introducing L75P/S153D/D321N mutations in pcDNA3-ERK2–3XFLAG plasmid [48]. All mutations thus generated were confirmed by Sangers sequencing. The primer list has been included in S9 Table.

## Generation of influenza A//WSN/1933 (H1N1) mutant NP plasmid constructs by Site Directed Mutagenesis (SDM)

Mammalian expression vector of influenza A//WSN/1933 (H1N1) NP protein -pCDNA6.2-NP-V5 and bacterial expression vector- pET28a-NΔ7NP-E339A expressing monomeric NP protein, were subjected to site-directed mutagenesis to generate NP S450A, T472A, S473A, S450A/ S473A, T472A/S473A and S450A/T472A/S473A mutations, using the QuikChange II Site-directed mutagenesis kit (Agilent Technologies) according to the manufacturer's instructions, further verified using Sangers DNA sequencing.

## PKCα knock down cell line generation

PKCα knock down cell line was generated by using mission shRNA (Sigma) specifically targeting *PRKCA* gene in A549 cells. HEK293T cells were transfected with plasmids containing either (a) pLKO.1-puro shRNAs (TRCN0000001690) targeting *PRKCA* (CTTTGGAGTTTCGGAGCTGAT) (shPKCα-1) or (b) pLKO.1-puro shRNAs (TRCN0000001691) targeting *PRKCA* (CGAGCTATTTCAGTCTATCAT) (shPKCα-2) or (c) a non-targeting control pLKO.1-puro plasmid DNA (NTC), SHC002, along with psPAX2 (Addgene plasmid #12260) and pVSVG (Addgene plasmid #138479). Lentiviral particles thus produced were either used alone or in combination to transduce A549 cells. Transduced cells expressing shPKCα-1, shPKCα-2 and shPKCα-1+2 thus produced were subjected to puromycin (5 µg/ml) (Sigma) selection. *PRKCA* gene knockdown was confirmed by western blot analysis of the cell lysate.

## Generation of recombinant NP mutant viruses

The mutations S450A, S473A and S450A/S473A were introduced in to the bidirectional pBD-NP plasmid DNA by SDM. Rescue of wild type and recombinant viruses were performed by using the influenza bidirectional reverse genetics plasmid system (pBD). The virus rescue constructs pBD-PB2, pBD*-PB1, pBD-PA and pBD-NP express both negative sense viral RNA (vRNA) and positive sense mRNA and have been used to express polymerase subunits, NP proteins and corresponding genomic RNA segments as described previously by Hoffmann et al., 2000 [57,58]. pTMΔRNP derived from pTM-polI-WSN-All [59] was created by excising the RNP expression cassette and was used for providing other viral genomic RNA segments (HA, NA, M and NS respectively) [58,60]. Briefly, 1.5 ug pTMΔRNP plasmid along with 0.25 ug each of pBD-PB2, pBD*-PB1, pBD-PA and pBD-NP (A/WSN/1933 (H1N1) WT or NP mutants) plasmids were transfected into a co-culture having $1.5 \times 10^6$ HEK293T and $0.5 \times 10^6$ MDCK cells using Lipofectamine 3000 according to the manufacturer's instructions. 24 h post transfection the initial medium was changed to virus growth medium (VGM: DMEM containing 0.2% bovine serum albumin (BSA) and 25 mM HEPES buffer) [36] supplemented with 0.5 µg/ml TPCK-treated trypsin and the supernatant (Passage 0 or P0) was harvested at 48 h.p.t. Viral titers were determined by standard plaque assay [36]. Subsequently amplified through two consecutive passaging (P2) and the cDNA of the P2 virus was subjected to Sangers sequencing. The P2 virus was utilized for subsequent characterization experiments. The bidirectional plasmids were obtained as a kind gift from Prof. Andrew Mehle, University of Wisconsin Madison, USA.

## Multicycle replication assay

Influenza A/WSN/1933 (H1N1) virus multicycle replication kinetics was performed following the protocol described by Mondal et al., 2017 [36]. MDCK cells were infected either with WT A/WSN/1933 (H1N1) virus or viruses harbouring phosphonull NP mutations (S450A, S473A, S450A/S473A) at a MOI of 0.01. For assessing the role of PKCα upon virus life cycle, control or PKCα-knock down stable A549 cells were infected with either A/WSN/1933 (H1N1) WT or mutant NP virus (S450A, S473A and S450A/S473A respectively) at a MOI of 0.01. Alternatively, A549 cells transiently expressing pcDNA3 or PKCα-DN were subsequently infected with A/WSN/1933 (H1N1) viruses at a MOI of 0.01 at 24 hours post transfection. Supernatant containing virus particles were collected at different time points and subsequently the viral titers were determined by performing plaque assay on MDCK cells [36].

**In vitro kinase assay**

The In vitro kinase assay of ERK2 and MEK1/2 was performed according to the protocol described by Ludwig et al., 1996 [61,62]. Briefly, HEK293T cells were transfected with FLAG tagged ERK2, MEK1 and MEK2 plasmids respectively for 23 hrs. The cells were left unstimulated or stimulated with PMA for 1 hr followed by lysis (for ERK2) in modified RIPA buffer (25 mM Tris [pH 7.5], 137 mM NaCl, 0.1% SDS, 0.5% Deoxycholate, 1% Nonidet P 40 (NP-40), 2 mM EDTA, 10% glycerol, protease and phosphatase inhibitors) and immunoprecipitation was carried out using FLAG antibody. Immune complexes were captured with Protein A magnetic beads (Biorad), which were then washed once in Triton lysis buffer (TLB; 20 mM Tris–HCl pH 7.4, 500 mM NaCl, 10% glycerol, 1% Triton X-100, 2 mM EDTA, protease and phosphatase inhibitor) and then twice in ERK2 kinase buffer ($MgCl_2$ 10 mM, 25 mM HEPES pH 7.5, 0.5 mM DTT, protease and phosphatase inhibitors) and finally resuspended in the same buffer. Influenza A/WSN/1933 (H1N1) wild type (WT) NP, oligomerization defective NP E339A or NP E339A harboring different phosphosite mutants (as described in specific figures) were purified from bacteria following the existing protocol and treated with RNaseA for 1 hr prior to be used as substrate [36]. 4 ug of the NP protein was incubated with protein-A magnetic sure beads (Biorad) containing immuno-captured kinases in the presence of 10 mCi of γ-32P ATP and 5 mM of cold ATP at 30°C for 45 min with intermittent shaking in every 10 minutes. Reactions were terminated by boiling in 1X Laemmli buffer, analyzed by SDS-PAGE, the gel was dried and subjected to phosphor imaging. Alternatively, the invitro phosphorylated (by ERK2) recombinant purified NP was subjected to LC- MS/MS analysis (outsourced from V Proteomics, India).

In vitro kinasing of bacterially expressed A/WSN/1933 (H1N1) NP E339A (or other mutants as described) was carried out using PKCδ CAT purified from HEK293T cells overexpressing the kinase through transient transfection following the protocol as described previously by Mondal et al., 2017 [36].

For analogue sensitive ERK2, in vitro kinasing was performed using 1mM γ-thio-ATP (ab138911, Abcam) or γ-thio-ATP-analogue (N6-Phenylethyl γ-thio-ATP, Biolog Life Science Institute, Bremen, Germany). The thio-phosphate conjugated substrates were alkylated using 2.5 mM of p-nitro benzyl mesylate (PNBM, ab138910, Abcam) at $22^0C$ for 2 hours with intermittent shaking every 15 minutes. Finally, the reaction was analysed through SDS-PAGE and detected by western blotting using anti-thiophosphate ester antibody (Abcam).

**In cell kinasing using analogue sensitive ERK2**

HEK293T cells transiently transfected with plasmids overexpressing ERK2 or ERK2-AS or empty vector control were infected with influenza A/ WSN/1933 (H1N1) virus or mock infected (MOI 0.5). At 12 hpi, cells were stimulated with 1.5 µg/ml PMA for 30 minutes or left unstimulated. Subsequently, $1.5 \times 10^6$ cells from each set were resuspended in ice cold ERK2 kinase buffer (mentioned above) reconstituted in PBS containing complete protease and phosphatase inhibitor cocktails along with 50 ug/ml digitonin (Sigma) and incubated on ice for 5 min. In cell kinasing was carried out in the same buffer (without digitonin) containing 100 µM N6-phenethyl ATP-γ-S and 1mM ATP at 30°C for 40 minutes with gentle rocking. Cells were then lysed in 0.5 ml RIPA buffer (50 mM Tris-HCl (pH 8), 150 mM NaCl, 1.0% NP-40 and 0.1% SDS) containing 25 mM EDTA followed by immunoprecipitation using anti-RNP antibody. The immunoprecipitated thio-phosphate conjugated NP was alkylated using 2.5 mM of p-nitro benzyl mesylate (PNBM) for 2 hrs at 22°C and detected through western blotting using anti-thiophosphate ester antibody.

**Immunofluorescence assay and image analysis**

A549 cells grown on coverslips were infected with A/ WSN/1933 (H1N1) virus at a MOI of 5 followed by fixation with 3% formaldehyde at different times post infection as indicated. Fixative was quenched with 0.1 M Glycine and the cells were permeabilized with 0.1% Triton-X 100 in PBS for 15 min at room temperature, followed by blocking with 3% BSA in PBS for an hour in room temperature. Cells were incubated with respective primary antibodies in blocking solution for overnight at 4°C. NP was detected with anti-RNP primary antibody and Alexa Fluor 488-conjugated donkey anti-goat IgG secondary antibody (I:1000

dilution) (Invitrogen). Subsequently, ERK2, MEK1 and PKCα were detected with specific primary antibodies at recommended dilution (CST) and Alexa Fluor 568, Alexa Fluor 555-conjugated donkey anti-rabbit IgG and anti-mouse secondary antibody (1:1000 dilution) (Invitrogen) respectively. DAPI (1 µg/ml) was used to stain the nucleus. Cells were imaged using 63x and 100x oil immersion objective lenses at 405 nm, 488 nm and 561 nm lasers in confocal microscope (Olympus FV3000) and post-processed with ImageJ software. 16-bit raw data sets were thresholded and converted to TIFF stacks for analysis.

Specific signal intensity of NP in the nucleus and throughout the cell was measured using ImageJ software. 50 cells (n = 50) were randomly chosen for each experimental set (at least 10 cells per field). Nucleus/ total cell intensity was calculated for each cell and plotted on the Y axis of the graph. Mean intensity with standard deviation of each set was compared by one-way ANOVA. Each image is a representative of three independent biological replicates.

### Super resolution structured illumination microscopy (SIM) and image analysis

A549 cells grown on coverslips were infected with A//WSN/1933 (H1N1) virus at a MOI of 5 following protocol as mentioned above in IFA. Super resolution imaging was performed on an Elyra 7 lattice SIM imaging system (ZEISS) equipped with 63x oil immersion objective, and 405 nm, 488 nm and 561 nm diode lasers. Raw data was acquired and reconstituted as per the standard protocol using appropriate oil refractive index to generate a supper resolution Lattice SIM$^2$ image (resolution excellence down to less than 60 nm) in the ZEN software (ZEISS). 32-bit reconstituted data sets were thresholded and converted to TIFF stacks for analysis.

Colocalization of NP – ERK2/MEK1/PKCα was analysed in image J (Fiji) software. Region of interest was selected with a rectangle tool and RGB profile plot was generated. Pearson correlation coefficient and Mander's overlap coefficients M1 and M2 were calculated for the total field for each data sets.

### Immunoprecipitation assay

HEK293T cells expressing A/ WSN/1933 (H1N1) virus NP and other interacting partners (ERK2, MEK1 and PKCα-CAT, PKCα-CAT-DN or full length PKCα WT(PKCα-FL) were lysed in radio-immunoprecipitation assay (RIPA) buffer (50 mM Tris-HCl (pH 7.5), 150 mM NaCl, 2 mM EDTA, 1% NP-40, 0.5% deoxycholate, 0.1% SDS) supplemented with 5 mg/ml of BSA and clarified by centrifugation as described [36]. Lysates were incubated with appropriate antibodies and immunocomplexes were captured on Protein A sure beads (Biorad). Beads were subsequently washed once with RIPA buffer containing 500 mM NaCl and 5 mg/ml BSA and finally twice in RIPA buffer without BSA. Immunoprecipitated samples were analyzed by western blotting.

A549 cells were infected with either A/WSN/1933 (H1N1) or B/Brisbane/60/ 2008 virus (MOI 10) respectively. The infected cell lysates were subsequently proceeded for immunoprecipitation assay. Lysates were incubated with NP antibody (A-NP H16L10 antibody, BEI Resources or B-NP monoclonal antibody, GT371, Invitrogen) and subsequently captured using Easy view beads (Sigma). The beads were washed with RIPA buffer and finally the immunoprecipitated complexes were analysed by western blot assay.

### LC-MS analysis

A549 cells were infected with influenza A virus strains, A/ WSN/1933 (H1N1) encoding PB2 with a C-terminal Flag (WSN-PB2-FLAG) (MOI 10) and at 24 hpi, the cells were lysed in RIPA buffer. RNP complexes were purified using anti-FLAG affinity resin (Sigma) followed by resolving the complex through SDS-PAGE and further processed by Coomassie blue staining. The A/ WSN/1933 (H1N1) virus NP protein band was excised from the gel, cut into small pieces (~1mm$^2$) and subjected to in-gel trypsin digestion protocol. The eluted peptides were utilized for mass spectrometric analysis. Peptides were characterized using a Thermo Q-exactive-HF-X mass spectrometer coupled to a Thermo Easy nLC 1200. Samples separated at 300nl/min on an Acclaim PEPMAP 100 trap (75 uM, 2 CM, c18 3 um, 100A) and an easyspray 100 Column (75 um, 25 cm, c18, 100A) using a 120 minute gradient with an initial starting condition of 2% B buffer (0.1% formic acid

in 90% Acetonitrile) and 98% A buffer (0.1% formic acid in water). Buffer B was increased to 28% over 90 minutes, then up to 40% in an additional 10 minutes. High B (90%) was run for 15 minutes afterwards. The mass spectrometer was outfitted with a Thermo nanospray easy source with the following parameters: Spray voltage: 1.8, Capillary temperature: 250°C, Funnel RF level=40. Parameters for data acquisition were as follows: for MS data the resolution was 60,000 with an AGC target of 3e6 and a max IT time of 50 ms, the range was set to 400–1600 m/z. MS/MS data was acquired with a resolution of 15,000, an AGC of 1e5, max IT of 50 ms, and the top 30 peaks were picked with an isolation window of 1.6 m/z with a dynamic execution of 25 s.

### MS data analysis

The resulting samples were processed using Thermo Proteome Discoverer 2.20.388. The custom database was made with supplied sequences that were downloaded from UniProt and searched with the following parameters: a tryptic enzyme with a max of 2 missed cleavages, a precursor mass tolerance of 10 ppm, and a fragment mass tolerance of 0.02 Da with pSTY modifications. The FDR rate was set at 0.01.

### Proximity Ligation Assay (PLA)

Approximately $6 \times 10^4$ A549 cells grown on eight well chamber glass slides were infected with influenza A virus strain, A/WSN/1933 (H1N1) (MOI 5). The cells were fixed with 4% Paraformaldehyde for 20 min and permeabilized with 0.2% Triton X-100 in PBS for 5 min. PLA was performed according to the manufacturer's instructions using the following kits and reagents: Duolink In Situ PLA Probe Anti-Rabbit MINUS (Sigma-Aldrich # DUO92005), Duolink In Situ PLA Probe Anti- Goat PLUS (Sigma-Aldrich # DUO92003), Duolink In Situ Detection Reagents Red (Sigma-Aldrich # DUO92008). The cells were probed with primary ERK2 antibody raised in rabbit (Cell signaling) and RNP antibody raised in goat (BEI resources) at 37°C for 1 h. For negative control, respective isotype matched IgG antibodies were used. Post ligation, rolling circle replication was carried out for 85 min at 37°C. Cells were washed twice with Wash Buffer B for 10 min and the samples were processed for IFA using the protocol as mentioned above using anti NP antibody (H16/L10) raised in mouse (BEI resources). PLA signals were detected as distinct fluorescent dots or puncta at 640nm wavelength using Olympus FV3000 confocal microscope at 63x oil immersion objective lens. 16-bit raw data sets were thresholded and converted to TIFF stacks for analysis using imageJ analysis software provided by the respective manufacturers. Number of PLA dots were counted for 100 cells by ImageJ software for each experimental set and plotted on the Y-axis of the graph. Average number of dots with standard deviation of each set was compared by one-way ANOVA. Intensity of the PLA and NP signal, in the nucleus and throughout the cell were measured by ImageJ. Intensity ratio (Nucleus/ Nucleus +Cytoplasm) was plotted to show their colocalization. Mean intensity with standard deviation of each sets were compared by two-way ANOVA.

### Protein expression, purification and oligomerization state determination

A/WSN/1933 (H1N1) virus WT or mutant NP genes cloned in pET28a vector were expressed in *E. coli* strain BL21 (DE3) (Novagen) and purified using Ni-NTA affinity chromatography (Qiagen). Purified proteins were treated with RNaseA and further purified through a HiTrap *Heparin* HP column (GE Healthcare). Proteins were concentrated to 4 mg/ml and incubated at 4°C for 96 hours in buffer containing 50 mM Tris, pH 7.5, 200 mM NaCl and 1 mM TCEP. Subsequently, the oligomerization state of the proteins were analysed through size-exclusion chromatography using SEC 650 10 x 300 Column (Biorad) calibrated with appropriate molecular weight standards.

### RNA extraction and qRT-PCR

A549 cells were infected with A/WSN/1933 (H1N1) WT or mutant viruses at a MOI of 0.05. Total RNA was isolated from appropriately treated cells using the Trizol reagent (Invitrogen) following manufacturer's instructions. 3 µg RNA was reverse transcribed using M-MuLV reverse transcriptase according to the manufacturer's instructions (New England

Biolabs). For qPCR, 2 ul of cDNA was used as a template with iTaq Universal SYBR Green Supermix (Bio-Rad) according to the manufacturer's instructions. qPCR analysis was performed in QuantStudio5 Applied Biosystems real-time PCR using the A/WSN/1933 (H1N1) NP segment specific primers either targeting the negative sense genomic RNA (vRNA) or mRNA. All RNA levels were normalized to GAPDH mRNA levels and calculated as the delta-delta threshold cycle (ΔΔCT).

### Polymerase activity assay

HEK293T cells were transfected with pCMV-PA, pCMV-PB1, pCDNA-PB2-3xFLAG, pcDNA-NP-V5 (WT or mutant) protein expression plasmids and pHH21vNA-Luc or pHH21vNA-Luc reporter RNA plasmids using Lipofectamine-3000 according to the manufacturer's instructions (Invitrogen). Cells were lysed 24 hours post-transfection in cell culture lysis reagent (CCLR) (Promega), and luciferase activity was measured using the luciferase assay system luminometer -Glomax 20/20 (Promega). Expression of A/WSN/1933 (H1N1) RNP components were analyzed by Western blotting.

### Subcellular fractionation and gene expression analysis

Subcellular fractionation was carried out according to the protocol performed by Li et al.,2025 [63]. Briefly, a synchronised infection was performed in A549 cells using either WT influenza A/ WSN/1933 (H1N1) or with viruses harbouring phospho-null NP mutations (S450A/S473A) at a MOI of 1. Cells were washed twice in ice cold PBS followed by incubation in hypotonic buffer (50 mM Tris (pH 7.4), 10 mM KCl, 350 mM sucrose, 1 mM EDTA, 1 mM DTT, 0.1% Triton X-100 and 100 U/ml RNase Inhibitor (Thermo Fisher Scientific)] on ice with frequent vortexing for 10 min. After centrifugation, the supernatant was collected as the cytoplasmic fraction (CF). An additional washing of the nuclear pellet with hypotonic buffer was performed, which was finally resuspended in nuclear pellet lysis buffer [10 mM Tris (pH 7.0), 100 mM KCl, 5 mM $MgCl_2$, 0.5% NP-40, 10 μM DTT, and 100 U/ml RNase Inhibitor] to prepare the nuclear fraction (NF).

Total RNA was isolated from cells by using RNAiso Plus reagent (Takara) and 300 ng RNA was reverse transcribed using M-MuLV reverse transcriptase according to the manufacturer's instructions (New England Biolabs). In order to perform qPCR of respective genes, corresponding cDNAs were used as a template with iTaq Universal SYBR Green Supermix (Bio-Rad) according to the manufacturer's instructions. qPCR analysis was performed in QuantStudio5 Applied Biosystems real-time PCR using specific primers targeting either the A/WSN/1933 (H1N1) NP segment negative sense genomic RNA (vRNA) or host specific genes - GAPDH, nuclear marker (U2 snRNA) and cytoplasmic marker (12s rRNA). The primer list used in qPCR have is included in S9 Table. vRNA and GAPDH abundance (CT values) in the nuclear and cytoplasmic fractions were normalised against U2 snRNA and 12s rRNA, respectively (using ΔΔCT method), to calculate the relative abundance in the respective sub-cellular compartments.

### Statistical analysis

All experiments were performed in triplicates, and each data was repeated at least three times independently. Graphs were plotted using Graphpad Prism 10.1.2 and represented as mean standard deviations (n = 3). Results were compared by performing a two-tailed Student's t-test and also by using either one-way ANOVA or two-way ANOVA for the experimental sets accordingly. Significance was defined as P < 0.05, and statistical significance is indicated with an asterisk (*). *P < 0.001 were considered statistically significant.

Significance was denoted as, (ns > 0.05; *P ≤ 0.05; **P ≤ 0.01; ***P ≤ 0.001) and were considered to be statistically significant.

### Supporting information

**S1 Fig. (A) HEK293T cells either mock treated or treated with PMA or overexpressing PKCα-CAT were analyzed for ERK2 activation by western blot analysis using ERK2 or phospho-ERK2 antibody.** (B) Cells overexpressing NP-V5 with PKCα-CAT along with increasing concentration of ERK2-FLAG were analyzed by western blot analysis using

anti-V5 antibody. Band intensities of hyperphosphorylated and unphosphorylated NP were assessed from three independent experiments are plotted using image J software.
(TIFF)

**S2 Fig. In vitro phosphorylated NP was subjected to LC-MS/MS analysis.** (A) Table showing peptide coverage for NP with phosphorylation of specific amino acid residues. (B-D) showing chromatograms of the phsopho-peptides subjected to CID fragmentation. Y and B ions are labelled in blue and red respectively.
(TIFF)

**S3 Fig. NP purified from A/WSN/1933(H1N1) virus infected cells was subjected to LC-MS/MS analysis.** (A) Table showing peptide coverage for NP with phosphorylation of specific amino acid residues. (B-D) showing chromatograms of the phsopho-peptides subjected to CID fragmentation. Y and B ions are labelled in blue and red respectively.
(TIFF)

**S4 Fig. (A and B) WT and mutant NP proteins, expressed and purified from bacteria were subjected to extensive RNaseA treatment followed by size exclusion analysis to evaluate their oligomerization potential.**
(TIFF)

**S5 Fig. WT and mutant NP proteins expressed in A549 cells were fixed and immuno-stained with anti-NP antibody at 12 (A) and 24 (B) hours post transfection.** Cells were imaged using confocal microscope.
(TIFF)

**S6 Fig. (A) WT and S450A/ S473A mutant viruses were used to infect A549 cells.** RNPs were visualised at 2 and 8 hpi by staining with NP and PB2 specific antibodies. Image analysis: 50 cells from 5 different fields were analysed using image J software to present nuclear-cytoplasmic distributions of NP. Each image is a representative of three independent biological replicates. Two way Anova was used to measure the statistical significance between the individual sets with P value (ns > 0.05; *$P \le 0.05$; **$P \le 0.01$; ***$P \le 0.001$). (B) A549 cells infected with WT or S450A/ S473A mutant viruses (MOI: 2) were harvested at 2 and 8 hpi followed by nuclear-cytoplasmic fractionation and total RNA isolation. qRT-PCR was performed to analyze the abundance of viral RNA (vRNA) and GAPDH in the subcellular compartments through normalization against U2 snRNA and 12s rRNA as nuclear and cytoplasmic reference genes.
(TIFF)

**S7 Fig. Co-immunoprecipitation of A/WSN/1933(H1N1) NP with ERK2-FLAG.** HEK293T cells were transiently transfected to overexpress NP-V5 either alone or in the context of ERK2-FLAG and PKCα-CAT-HA overexpression and immunoprecipitated using FLAG antibody. Co-precipitation of NP was observed using V5 antibody. Co-precipitation of ERK2 was observed using FLAG and HA (PKCα) antibody.
(TIFF)

**S8 Fig. A549 cells were infected with influenza A/H1N1/WSN virus (MOI 10) followed by harvesting them different time points post infection.** ERK2 activation was monitored using phospho-ERK antibody. NP served as infection marker while GAPDH as loading control.
(TIFF)

**S9 Fig. Extent of co-localization of PKC, ERK and MEK (in red line) with NP (in green line) was analyzed from the super-resolution micrographs (area specified by white rectangle) presented in Fig 7D-7F and plotted using image J software.**
(TIFF)

**S10 Fig. Mock and A/WSN/1933 (H1N1) infected [at 2 hours (A) or 6 hours (B) of post infection] cells were subjected to PLA using anti-RNP (goat) and anti-ERK2 (rabbit) primary antibodies or with one of the primary**

**antibodies along with the IgG isotype control for the other.** Post PLA, cells are stained with anti-NP antibody (mouse) and imaged. Quantitative analysis of the PLA dots are presented in main text Fig 8D-8F.
(TIFF)

**S1 Table. Ion series for the peptide – LMESARPEDVSF.**
(DOCX)

**S2 Table. Ion series for the peptide: LMESARPEDVSFQGR.**
(DOCX)

**S3 Table. Ion series for the peptide: GVFELSDEKATSPIVPSFDMSNEGSY.**
(DOCX)

**S4 Table. Ion series for the peptide: ATSPIVPSFDMSNEGSYFFGDNAEEYDN, T2-Phospho (79.96633 Da) Charge: +3, Monoisotopic m/z: 1061.75223 Da.**
(DOCX)

**S5 Table. Ion series for the peptide: ATSPIVPSFDMSNEGSYFFGDNAEEYDN, S3-Phospho (79.96633 Da).**
(DOCX)

**S6 Table. Ion series for the peptide: LMESARPEDVSFQGR, S4-Phospho (79.96633 Da).**
(DOCX)

**S7 Table. Percentage conservation of the ERK2 phosphorylation sites in different influenza virus NP proteins.**
(DOCX)

**S8 Table. Coefficient of Co-localization from super resolution microscopy of PKC, MEK1 and ERK2 (in red) with NP (in green).**
(DOCX)

**S9 Table. List of primers.**
(DOCX)

**S1 Data. Source data for Figs 1, 2, 3, 5, 6, 8, 9, S1, S7, and S8.**
(DOCX)

## Acknowledgments

We acknowledge the Confocal Microscope purchased under DST-FIST grant conferred on the Department of Bioscience and Biotechnology, Indian Institute of Technology Kharagpur, India, File no. SR/FST/LS-I/2019/595(C). We also thank Ms. Elizabeth Remily-Wood, Director of the USF MCOM Proteomics facility. The proteomics work has been supported, in part, by the Morsani College of Medicine at the University of South Florida.

## Author contributions

**Conceptualization:** Arindam Mondal.

**Data curation:** Indrani Das Jana, Soumik Dey, Manoj Si, Arunava Roy.

**Formal analysis:** Indrani Das Jana, Soumik Dey, Arunava Roy, Arindam Mondal.

**Funding acquisition:** Arunava Roy, Arindam Mondal.

**Investigation:** Indrani Das Jana, Soumik Dey, Manoj Si, Arunava Roy, Arindam Mondal.

**Methodology:** Indrani Das Jana, Soumik Dey, Manoj Si, Arunava Roy, Arindam Mondal.

**Project administration:** Arindam Mondal.

**Resources:** Arindam Mondal.

**Supervision:** Arindam Mondal.

**Writing – original draft:** Indrani Das Jana, Arindam Mondal.

**Writing – review & editing:** Indrani Das Jana, Soumik Dey, Arindam Mondal.

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
