## [Decision Letter · Decision Letter 0]

23 Jul 2025

Host Protein Kinase C⍺: the master regulator of the MAPK mediated nuclear export of influenza virus ribonucleoprotein complexes.

PLOS Pathogens

Dear Dr. Mondal,

Thank you for submitting your manuscript to PLOS Pathogens. After careful consideration, we feel that it has merit but does not fully meet PLOS Pathogens's publication criteria as it currently stands. Therefore, we invite you to submit a revised version of the manuscript that addresses the points raised during the review process.

Please submit your revised manuscript within 60 days Sep 21 2025 11:59PM. If you will need more time than this to complete your revisions, please reply to this message or contact the journal office at plospathogens@plos.org. Please include the following items when submitting your revised manuscript:

We look forward to receiving your revised manuscript.

Kind regards,

Luis Martínez-Sobrido

Academic Editor

PLOS Pathogens

Kanta Subbarao

Section Editor

PLOS Pathogens

Editor-in-Chief

PLOS Pathogens

orcid.org/0000-0003-2946-9497

Editor-in-Chief

PLOS Pathogens

orcid.org/0000-0002-7699-2064

**Journal Requirements:**

- TM on page: 32.

Potential Copyright Issues:

i) Figures 1A, 1G, 8D, and Graphical Abstract. Please confirm whether you drew the images / clip-art within the figure panels by hand. If you did not draw the images, please provide (a) a link to the source of the images or icons and their license / terms of use; or (b) written permission from the copyright holder to publish the images or icons under our CC BY 4.0 license. Alternatively, you may replace the images with open source alternatives. See these open source resources you may use to replace images / clip-art:

5) We note that you have included Graphical Abstract in the manuscript file. Please upload it as a separate Figure file in .tif or .eps format. For more information about how to convert and format your figure files please see our guidelines:

https://journals.plos.org/plospathogens/s/figures#loc-file-requirements  

**Reviewers' Comments:**

Reviewer's Responses to Questions

**Part I - Summary**

Reviewer #1: The manuscript by Das Jana et al provides relevant information about the interaction of host Kinase and the pathway involving ERK2, PKC alfa and other factors in the context of influenza, specifically through the interaction with NP. The work presented is scientifically sound and uses a combination of multiple techniques to delineate how ERK2 interacts with NP mechanistically, specific molecular signatures involved, how other host factors are or are not involved and the role of these interactions in the context of viral infections with influenza A but also influenza B. This is a well-written manuscript providing novel mechanistic data of the viral-host interactions of influenza that is relevant for the readers of PLOS Pathogens. Overall, I have outlined both significant and minor comments below that I would like the authors to address.

Major comments

The significant number of Western blots presented requires the inclusion of all whole membranes as supplementary material, which is also critical for the peer-review process.

Overall, the quality of the figures is poor. I made a significant effort to review this paper given the quality of the provided figures. I do not know if the issue relates to the original quality of the materials or the document generation for revision. Although I was able to capture most of the information, I cannot assess the IFI results presented in Figures 4, 7, and 9. A revised version should be submitted with higher-quality figures to facilitate a proper assessment of the results presented in the aforementioned figures.

Minor comments

Line 327:…”Similar to ERK2, MEK1 but not MEK2 interacted with NP in a PKCα dependent manner”… Can the differences be explained by the total NP protein, clearly different between MEK1 and MEK2 experiments?

The results presented about the role of ERK2 in the distribution of NP are impossible to assess due to the low quality of the pictures.

Can the limitation imposed by NP-Changes be overcome after passages of the virus?

E339A role? Has it been addressed in the context of whole virus?

”Viral mRNA, cRNA and vRNA synthesis was significantly boosted for RNPs reconstituted with specific phosphomimetic variants of NPs (Fig 3I)”... It’s impossible to agree since comparisons with carrying A are not present.

Figure 1E shows a specific involvement of ERK2 in the phosphorylation of NP. However, the Western blots presented do not allow a firm conclusion about the expression of ERK2. Similar bands are observed when MEK1 is overexpressed, which should not be the case.

Results presented in figure 2H are hard to put into context due to the lack of NP wt (like in line 2 in figure 2G) and a condition without PKCδ. It is important to assess whether the detrimental effect observed in the presence of the mutant in figure 2G is reproducible for the experiments conducted in Fig 2H.

How is the level of replication between WT and the mutants for Figure 9? As presented, the results are misleading, as it was previously established that mutant replicates with a lower efficiency, which is not apparent in a graph normalized to 100%.

Please provide the primers used in the study.

The anti-RNP ab recognizes what exactly? Please provide more precise information. Is it an anti-PB2?

Line 248: The Authors mention a timeline for influenza infection that does not represent the experimental conditions used. The cited paper is based on human infections, not on in vitro systems utilizing highly permissive cells and laboratory-adapted strains.

Line 175 and 176: The text refers to figure 1H and not 1F

Line 574: Spell out the strain A/PR/8/34

Line 575-576: Was PMA added on top or the whole media was replaced with media + PMA?

Line 628: Provide more background information about pTMΔRNP plasmid and references.

Line 631: Provide composition of VGM.

Line 653: Please modify Invitro to in vitro

Reviewer #2: In this manuscript, the authors explore the mechanism of influenza A virus regulation by PKC alpha and the MAP kinase cascade, particularly NP export from the nucleus. The fact that PKC alpha and MAPK regulate replication was previously known, but the molecular players in the pathway had not yet been elucidated. Through in vitro and cell-based experiments, including both transfected and infected cells, the authors show that ERK2 is directly responsible for phosphorylating NP, but PKC alpha is required to activate ERK2 and also to act as a scaffold to bring ERK2, MEK1 and NP together. Moreover, they show that mutants that cannot be phosphorylated replicate less well, and that this is consistent with retention of NP in the nucleus at later times post infection, when NP should be moving to the cytoplasm (presumably as part of vRNPs). Similarly, they show that loss of PKC alpha activity reduces replication and NP export. The findings are significant because they provide a molecular pathway from PKC alpha to NP that explains regulation of influenza replication by this kinase and its downstream effectors. The experiments are generally strong and rigorous. I mostly have comments related to the writing and presentation.

Reviewer #3: In their manuscript Jana and colleagues elucidate the role of protein kinase C alpha in influenza virus RNP export. The paper is interesting but there are several points that need the authors’ attention.

**Part II – Major Issues: Key Experiments Required for Acceptance**

Reviewer #1: The significant number of Western blots presented requires the inclusion of all whole membranes as supplementary material, which is also critical for the peer-review process.

Overall, the quality of the figures is poor. I made a significant effort to review this paper given the quality of the provided figures. I do not know if the issue relates to the original quality of the materials or the document generation for revision. Although I was able to capture most of the information, I cannot assess the IFI results presented in Figures 4, 7, and 9. A revised version should be submitted with higher-quality figures to facilitate a proper assessment of the results presented in the aforementioned figures.

Reviewer #2: 1. Lines 289-294 – this section is odd. What explains the change in localization of NP over time, in the absence of any stimulation? They have shown in Fig. 1 that NP phosphorylation in transfected cells requires PMA stimulation. Minimally, they should explain how they interpret this and why it is likely to be informative. Ideally, they should just show the data in infected cells, which is anyway more relevant. They may have to do additional experiments with the phosphomimetic mutants if they feel like this is an important point for their study.

Reviewer #3: Major points

1) The authors call PCKa a master regulator of the influenza virus life cycle. However, a virus lacking the appropriate phosphorylation sites grow as well as wild type. Since the virus seems to be fit without this whole interaction, the authors need to tone down their language.

2) Many parts of the manuscript are written in a relatively sloppy way (see below), especially the methods section. This reflects bad on the conduct of the science itself too.

3) It is unclear if the authors did their main experiments with A/WS/33 or A/WSN/33.

4) The figures seem to be at a very low resolution.

**Part III – Minor Issues: Editorial and Data Presentation Modifications**

Reviewer #1: Line 327:…”Similar to ERK2, MEK1 but not MEK2 interacted with NP in a PKCα dependent manner”… Can the differences be explained by the total NP protein, clearly different between MEK1 and MEK2 experiments?

The results presented about the role of ERK2 in the distribution of NP are impossible to assess due to the low quality of the pictures.

Can the limitation imposed by NP-Changes be overcome after passages of the virus?

E339A role? Has it been addressed in the context of whole virus?

”Viral mRNA, cRNA and vRNA synthesis was significantly boosted for RNPs reconstituted with specific phosphomimetic variants of NPs (Fig 3I)”... It’s impossible to agree since comparisons with carrying A are not present.

Figure 1E shows a specific involvement of ERK2 in the phosphorylation of NP. However, the Western blots presented do not allow a firm conclusion about the expression of ERK2. Similar bands are observed when MEK1 is overexpressed, which should not be the case.

Results presented in figure 2H are hard to put into context due to the lack of NP wt (like in line 2 in figure 2G) and a condition without PKCδ. It is important to assess whether the detrimental effect observed in the presence of the mutant in figure 2G is reproducible for the experiments conducted in Fig 2H.

How is the level of replication between WT and the mutants for Figure 9? As presented, the results are misleading, as it was previously established that mutant replicates with a lower efficiency, which is not apparent in a graph normalized to 100%.

Please provide the primers used in the study.

The anti-RNP ab recognizes what exactly? Please provide more precise information. Is it an anti-PB2?

Line 248: The Authors mention a timeline for influenza infection that does not represent the experimental conditions used. The cited paper is based on human infections, not on in vitro systems utilizing highly permissive cells and laboratory-adapted strains.

Line 175 and 176: The text refers to figure 1H and not 1F

Line 574: Spell out the strain A/PR/8/34

Line 575-576: Was PMA added on top or the whole media was replaced with media + PMA?

Line 628: Provide more background information about pTMΔRNP plasmid and references.

Line 631: Provide composition of VGM.

Line 653: Please modify Invitro to in vitro

Reviewer #2: 2. The author summary is too technical.

3. In Fig. 1H, they do detect some phosphorylation in the cells that are lacking ERK2-AS. Is that cross-reactivity from the antibody? They should address it in the text.

4. Line 205 – I assume they mean NP S473?

5. Fig. 2E – in the diagram, should it not be T472?

6. In Fig. 2G, the top panel (pNP) has some cropped lanes but the others do not seem to. First of all, the authors should make it more obvious if they cropped lanes (for example use a dashed line). Second, are these from the same experiment? Why were lane not cropped for the other proteins?

7. In the legend for Fig. 2D,E,F, they should specify the full name of the strains from which they got the NP sequences, not just the subtype.

8. Fig. 2 legend, line 1056 – they should specify that the plotted measurements are normalized to ERK2 phosphorylation of WT NP (alternatively, they can specify that in the axis label of Fig. 2A)

9. Line 232 – I am not sure I agree with the statement that the mutants have lower replication kinetics – they initially look the same, but never reach the same level. It is unclear if they ever would.

10. Line 244 – this sentence should be rephrased because it makes it sound like there is less of a reduction in vRNA synthesis at 48 hpi, as grammatically the 5% refers to reduction. What they mean is a 95% reduction.

11. Lines 258-260 – It is not clear what they think causes the positive regulation of RNA synthesis with the phosphomimetic mutants.

12. Fig, 3 – the graphs showing time points of replication/viral genome (panels B, D, and F) should be plotted in an actual linear scale – right now the x-axis is a categorical variable, with equal distances representing 5 hrs to 24 hrs.

13. Fig. 3 legend – Would be good if they specified the multiplicity of infection for assays in panels C to F.

14. Lines 302-303 – they should clarify how these data show that the events are independent of RSK1, I am not clear why.

15. Fig. 4C, right – it would be more informative to show significance against WT, instead of between the mutants

16. Fig. 4, 7, 9B legends – they should specify how many biological replicates of the microscopy experiments they did, because the reference to fields makes it sound like it was 1, which is not sufficient.

17. Line 307 – the sentence and section start with “To this end,” but it’s not clear what end they are talking about.

18. Line 350 – Should be “it failed to…”

19. In the diagrams in Fig. 6, the inhibitory arrows from NP/ERK2 to inactive PKC alpha are confusing because they suggest that NP/ERK2 are preventing something, whereas it is actually the lack of active PKC alpha that is preventing the interaction. They should find a different way to represent the missing interaction.

20. Throughout the manuscript, they should have a space between numbers and units (48 hpi, not 48hpi for example). Currently they have it some of the time but not all of the time.

21. Line 534 – they should rephrase “ERK downstream RSK1 mediated NP phosphorylation” because it is hard to follow.

22. Line 651, 653 – should be in vitro instead of invitro.

23. Line 712 – how did they define the nucleus vs. cytoplasmic compartments?

Reviewer #3: Minor points

1) Lines 33, 85, 108, 566, 567, 582 : There is no need to start words with capitalized letters mid-sentence.

2) Line 34: Please define Raf, MEK, ERK and MAPK.

3) Line 48: ‘a’, not ‘the’ – also see comment above about key regulator.

4) Lines 59, 145-147, 191-194 : Please rephrase these sentences.

5) Line 71: Please tone that down. Current antivirals work well against almost all strains. Vaccines work too, with caveats.

6) Line 77: ‘gets’, not ‘get’

7) Line 89: ‘phosphorylations’

8) Throughout the manuscript: Please define abbreviations when first used. This includes: Raf, MEK, ERK, HA, HEK, SDS-PAGE, ATP, MDCK, hpi, FBS, DMSO, UTRs, BNP

9) Line 97: ‘the MAPK pathway’

10) Line 98: ‘of the virus life cycle’

11) Line 114: ‘facilitates’

12) Line 121: ‘modulates’

13) Line 123: ‘phosphorylates’

14) Lines 125, 142, 157, 158, 160, 322, 381, 397, 401, 481, 571: Many different versions of the strain name A/WSN/1933 are used throughout the manuscript. Please stick to ‘A/WSN/1933 (H1N1)’.

15) Line 153: ‘an in vitro’

16) Line 159: ‘an extend’ not ‘the extend’

17) Line 163: ‘the virus life cycle’

18) Line 169: ‘MOI’, not ‘MO1’

19) Line 169: ‘with PMA’

20) Line 171: ‘with thio-phosphate’

21) Line 198: ‘does not’, not ‘doesn’t’

22) Line 238: ‘of the influenza virus’

23) Line 240: Remove ‘of’

24) Line 247: ‘showed a trend’

25) Line 251: ‘of the virus’

26) Line 257: ‘to a defect’

27) Line 274: ‘showing a minor’

28) Line 286: ‘phospho’ is misspelled.

29) Line 289: ‘a potential role’

30) Line 291: ‘render’

31) Line 294: ‘showed an intermediate’

32) Line 322: Please use proper strain names.

33) Line 331: ‘of the PB2’

34) Line 339: ‘mediating’

35) Line 343: Remove ‘the after ‘utilised’.

36) Line 348: ‘mutations’

37) Line 389: ‘the majority’

38) Line 398: ‘infected set’?

39) Line 400: Remove the comma after ‘if’.

40) Line 428: Remove ‘of’ after ‘times’.

41) Line 429: Please rephrase this sentence.

42) Line 475: ‘flight’ is not the right word here.

43) Line 478: ‘to the cytoplasm’

44) Line 486: ‘supporting the virus’

45) Line 499: Remove the ‘if’ after ‘that’.

46) Line 501: ‘of the MAPK cascade’

47) Line 502: ‘of the infection’

48) Line 535: Should be ‘promotes’ and ‘the vRNP’

49) Line 548: ‘the virion’

50) Line 558: ‘detailed’

51) Line 569: What is the concentration of antimycotic (and which one was used), penicillin and streptomycin?

52) Line 572: Is ‘FLAG tag’ meant?

53) Line 572: There is a space missing.

54) Line 585: ‘virus’

55) Line 586: There is a space missing.

56) Line 590: ‘the full-length’

57) This reviewer gave up pointing out issues after line 590 – but it did not get any better.

PLOS authors have the option to publish the peer review history of their article (what does this mean? ). If published, this will include your full peer review and any attached files.

**Do you want your identity to be public for this peer review?** For information about this choice, including consent withdrawal, please see our Privacy Policy .

Reviewer #1: **Yes: ** C. Joaquín Cáceres

Reviewer #2: No

Reviewer #3: No

**Figure resubmission:**

**Reproducibility:**



---

## [Decision Letter · Decision Letter 1]

15 Oct 2025

PPATHOGENS-D-25-01255R1

Host Protein Kinase C⍺: the novel Mitogen Activated Protein Kinase (MAPK) specific scaffold regulating nuclear export of influenza virus ribonucleoprotein complexes.

PLOS Pathogens

Dear Dr. Mondal,

Thank you for submitting your manuscript to PLOS Pathogens. After careful consideration, we feel that it has merit but does not fully meet PLOS Pathogens's publication criteria as it currently stands. Therefore, we invite you to submit a revised version of the manuscript that addresses the points raised during the review process.

Please submit your revised manuscript within 30 days Dec 14 2025 11:59PM. If you will need more time than this to complete your revisions, please reply to this message or contact the journal office at plospathogens@plos.org. Please include the following items when submitting your revised manuscript:

We look forward to receiving your revised manuscript.

Kind regards,

Luis Martínez-Sobrido

Academic Editor

PLOS Pathogens

Kanta Subbarao

Section Editor

PLOS Pathogens

Sumita Bhaduri-McIntosh

Editor-in-Chief

PLOS Pathogens

orcid.org/0000-0003-2946-9497

Michael Malim

Editor-in-Chief

PLOS Pathogens

orcid.org/0000-0002-7699-2064

**Journal Requirements:**

**Reviewers' Comments:**

Reviewer's Responses to Questions

**Part I - Summary**

Reviewer #4: In this revised manuscript, the authors have adequately addressed the previous reviewers' relatively minor concerns through revisions to the text and new experimental data. Overall, their results support their conclusions and the data are presented in an easily interpretable manner. The model schematics in multiple figures help the reader interpret the experimental design and results. Congrats on this interesting study.

Reviewer #5: In this article, novel influenza virus-host interactions are described. Specifically, the authors show that the ERK2 kinase phosphorylates the influenza NP protein, and they identify the NP residues being phosphorylated by ERK2. In addition, they show that the NP phosphorylation is essential for vRNP export from the nucleus to the cytoplasm, and for virus propagation at late times post-infection. Furthermore, they show that PKC⍺ activates the MAPK cascade (leading to ERK2 activation), and PKC⍺ participates in stable interactions with the upstream kinase, MEK1, with ERK2, and with the influenza NP, thereby forming a multiprotein complex that regulates ERK2 activation, and NP phosphorylation. The results will be interesting for virologists and cellular biologists in general. However, there are some points which should be considered before publication.

Reviewer #6: Using an in vitro kinase assay, the authors show that influenza NP is phosphorylated by ERK2. To decouple oligomerization from phosphorylation, they introduced a mutation that prevents NP oligomerization and confirmed that NP remains an ERK2 substrate. LC-MS/MS mapped three sites—S450, S472, and S473—with a couple resides conserved across influenza A and B subtypes and lineages. The authors then probed the functional role of these modifications, presenting evidence that NP phosphorylation may facilitate nuclear export. Finally, they demonstrate that NP interacts with ERK only when ERK is in its phosphorylated (active) form and describe a multikinase complex that mediates NP phosphorylation.

**Part II – Major Issues: Key Experiments Required for Acceptance**

Reviewer #4: (No Response)

Reviewer #5: - The authors analyze the subcellular localization of the influenza NP protein, and NP variants lacking its phosphorylation sites in influenza virus-infected cells, by fluorescence microscopy, and they infer from these experiments that the ERK2-mediated, NP phosphorylation facilitates the transport of the vRNPs from the nucleus to the cytoplasm at the late stages of the infection (Figure 4C). From my point of view, this is an overstated conclusion that needs further support, by, for example, analyzing the nuclear and cytoplasmic localization of viral genomic RNAs, that form part of the vRNPs. From the experiments shown in Figures 4A, 4B (in cells overexpressing the NP variants) and 4C (in infected cells), they should only conclude that the NP phosphorylation affects its subcellular localization at late time points.

- The levels of phospho-ERK should have been tested in influenza-infected cells, to confirm that ERK2 is activated in infected cells, and to show the activation kinetics at different times post-infection

Reviewer #6: As a revised manuscript, the authors provide a large collection of datasets with figures within figures. Please take a concise effort to determine what data should be kept in the main figures and what data could be illustrated in supplementary figures.

Refrain from the use of language such as “severely restricted” when describing a virus that still can reach titers of >10^4 pfu/ml.

**Part III – Minor Issues: Editorial and Data Presentation Modifications**

Reviewer #4: (No Response)

Reviewer #5: - According to my point of view, it is not clear the conclusion indicating that “Co-expression of PKCα-CAT, resulted in moderate hyperphosphorylation of NP, which gets significantly boosted in presence of ERK2, in a dose dependent manner” (lines 153-154). For example, if we compare ERK2 expression in lanes 4 and 5, the expression of ERK2 is clearly higher in lane 5 (Figure 1D), whereas the NP phosphorylation levels are similar.

- Why the authors did not test if ERK1 participates on NP phosphorylation? This point should be at least discussed

- In Figure 1H, to what protein corresponds the lower band detected after NP immunoprecipitation (in the NP-PO3S blot), as this lower band is detected both in non-infected and infected cells?

- Which is the antibody used to detect phospho-NP in figure 2B? While in the presence of ERK-2AS and the ATP analogue, an antibody specific for thiophosphate ester seems to have been used, in the presence of regular ATP, this antibody should not detect any band, according to my understanding. However, in lanes 2 and 3 (Figure 2B), in which the regular ATP was used for the reaction, clear phospho-NP bands are observed. This result should be further explained.

- In line 219, the words “to this end” make no sense to me. The authors should better clarify why they mutate the S450, T472 and S473 residues to phospho-null alanine, either individually or in combination, in the monomeric NP-E339A.

- In the virus-infected cells, the authors detect that NP is phosphorylated at residues S450, T472 and S473, however, they only generate the mutant viral strains NP-S450A, NP-S473A. Why the authors did not generate the T472A mutant virus? This point should be further explained.

-It is surprising that in cells knocked-down for PCK⍺ (the shPKC⍺-1 cells), the expression of PKC⍺ is not much higher than in the PCK⍺-2 and PKC⍺1+2 cells, and despite this, viral titers at 12 h vary greatly among the shPKC⍺-1 cells and the shPKC⍺-2 cells or the shPKC⍺-1+2 cells. Do the authors have an explanation for this?

- In the co-IP assays, I miss a control. For example, after immunoprecipitating ERK2, NP is co-immunoprecipitated in the presence of PKC⍺, however, a control in which is ERK is not expressed, but NP and PKC⍺ are expressed is needed to discard the possibility that NP binds inespecifically to the beads used for the immunoprecipitation.

- Similarly, in the PLA assays (Figures 8E and 8F), I miss a control using mock-infected cells and staining the cells with the anti-NP and anti-ERK antibodies.

Reviewer #6: Figure 3 B – Y-axis should start at 1.

PLOS authors have the option to publish the peer review history of their article (what does this mean? ). If published, this will include your full peer review and any attached files.

**Do you want your identity to be public for this peer review?** For information about this choice, including consent withdrawal, please see our Privacy Policy .

Reviewer #4: No

Reviewer #5: No

Reviewer #6: No

**Figure resubmission:**

After uploading your figures to PLOS’s NAAS tool - https://ngplosjournals.pagemajik.ai/artanalysis, NAAS will process the files provided and display the results in the "Uploaded Files" section of the page as the processing is complete. If the uploaded figures meet our requirements (or NAAS is able to fix the files to meet our requirements), the figure will be marked as "fixed" above. If NAAS is unable to fix the files, a red "failed" label will appear above. When NAAS has confirmed that the figure files meet our requirements, please download the file via the download option, and include these NAAS processed figure files when submitting your revised manuscript
---

## [Decision Letter · Decision Letter 2]

20 Dec 2025

Dear Dr Mondal,

We are pleased to inform you that your manuscript 'Host Protein Kinase C⍺: the novel Mitogen Activated Protein Kinase (MAPK) specific scaffold regulating nuclear export of influenza virus ribonucleoprotein complexes.' has been provisionally accepted for publication in PLOS Pathogens.

Best regards,

Luis Martínez-Sobrido

Academic Editor

PLOS Pathogens

Kanta Subbarao

Section Editor

PLOS Pathogens

Sumita Bhaduri-McIntosh

Editor-in-Chief

PLOS Pathogens

orcid.org/0000-0003-2946-9497

Michael Malim

Editor-in-Chief

PLOS Pathogens

orcid.org/0000-0002-7699-2064

Reviewer Comments (if any, and for reference):

Reviewer's Responses to Questions

**Part I - Summary**

Reviewer #5: The authors have replied to my previous concerns adequately

Reviewer #6: The authors have addressed my concerns.

**Part II – Major Issues: Key Experiments Required for Acceptance**

Reviewer #5: (No Response)

Reviewer #6: N/A

**Part III – Minor Issues: Editorial and Data Presentation Modifications**

Reviewer #5: The authors have properly addressed all my comments. I have only one very minor comment:

1. In Fig. S6A panels, “6 hpi” is written, whereas I think it should be 8 hpi (as stated in the text and figure legend)

Reviewer #6: N/A

PLOS authors have the option to publish the peer review history of their article (what does this mean? ). If published, this will include your full peer review and any attached files.

**Do you want your identity to be public for this peer review?** For information about this choice, including consent withdrawal, please see our Privacy Policy .

Reviewer #5: No

Reviewer #6: No

---

## [Editor Report · Acceptance letter]

Dear Dr Mondal,

We are delighted to inform you that your manuscript, "Host Protein Kinase C⍺: the novel Mitogen Activated Protein Kinase (MAPK) specific scaffold regulating nuclear export of influenza virus ribonucleoprotein complexes.," has been formally accepted for publication in PLOS Pathogens.

Best regards,

Sumita Bhaduri-McIntosh

Editor-in-Chief

PLOS Pathogens

orcid.org/0000-0003-2946-9497

Michael Malim

Editor-in-Chief

PLOS Pathogens

orcid.org/0000-0002-7699-2064